



# On the potential of a low-complexity model to decompose the temporal dynamics of soil erosion and sediment delivery

Francis Matthews[1,2], Panos Panagos[1], Arthur Fendrich[1,3,4], Gert Verstraeten[2]

[1] European Commission, Joint Research Centre (JRC), Ispra 21027, Italy

[2] KU Leuven, Department of Earth and Environmental Sciences, Celestijnenlaan 200e, Leuven 3001, Belgium

[3] Laboratoire des Sciences du Climat et de l'Environnement, CEA-CNRS-UVSQ-UPSACLAY, Gif sur Yvette 91190, France

[4] Université Paris-Saclay, INRAE, AgroParisTech, UMR SAD-APT, 91120, Palaiseau, France

*Correspondence to*: Francis Matthews (francis.matthews@kuleuven.be, fmatthews1381@gmail.com )

**Abstract.**

Testing and improving the capacity of soil erosion and sediment delivery models to simulate the intra-annual dynamics climatic

drivers and disturbances (e.g. vegetation clearcutting, tillage events, wildfires) is critical to understand the drivers of the system variability. In seasonally changing agricultural catchments, explicit temporal dynamics are typically neglected within many soil erosion modelling approaches, in favour of a focus on the long-term annual average as the predictive target. Here, we approach the trade-off between the need for model simplicity and temporally-dynamic predictions by testing the ability of a low-complexity, spatially distributed model (WaTEM/SEDEM), to decompose the 15-day dynamics of soil erosion and

sediment yield. A standardised parameterisation and implementation routine was applied to four well-studied catchments in North-West Europe with open-access validation data. Through the testing of several alternative model spatial and connectivity structures, including the addition of an empirical runoff coefficient, we show that a temporally-static calibration of transport capacity cannot adequately replicate the relative seasonal decoupling of gross (on-site) soil erosion and sediment delivery. Instead, embedding seasonality into the calibration routine significantly improved the model performance, revealing a negative

relationship between gross (pixel-scale soil displacement) and net erosion (stream channel sediment load) throughout the year. By incorporating temporal dynamics, the relative net effect is a reduction in the magnitudes of the spatially-distributed sediment fluxes at aggregated timescales, compared to a temporally-lumped approach. Published catchment observations infer that the efficacy of sediment delivery via overland flow is strongly reduced in the summer by abundant vegetative boundaries and increased in the winter via soil crusting its promotion of runoff. Models operating at temporally-aggregated timescales

should account for the possibility of decoupling in time and space between gross erosion and sediment delivery on arable catchment systems, related to alternations between transport- and detachment-limited sediment transport capacity states. Despite the complexities involved in the temporal downscaling of WaTEM/SEDEM, we show the utility of this approach to: 1) identify key missing information components requiring attention to reduce error in gross erosion predictions (e.g. more consideration of antecedent soil conditions), 2) form a basis for strategically adding physical process-representation, with a





focus on maintaining low model complexity, and 3) better understand the spatial and temporal interdependencies within soil

erosion models when undertaking upscaling exercises.

## 1 Introduction

Soil erosion by water is gaining attention as a critical risk factor in a plethora of threats to soil health (Borrelli et al., 2022).

Landscape disturbances, typically through instances of tillage, forest clearcutting, or wildfires, have increased soil erosion

rates in the order of 1-2 orders of magnitude in some cases (Montgomery, 2007; Evans et al., 2020; Vieira et al., 2015). The

effects of increased soil erosion and redistribution rates are a key defining feature of the Anthropocene, in which humans have

caused a large-scale reconfiguration of the Earth's sediment, hydrological and biogeochemical cycles (Syvitski et al., 2005,

2022). The consequence is reduced soil lifespans (Evans et al., 2020), and a range of economic, ecological and societal

consequences (e.g. eutrophication, muddy floods, reservoir siltation, infrastructure damage) when eroded soils and bound

agricultural chemicals are redistributed in the landscape (Patault et al., 2021).

For the purpose of the first prediction-based management decisions, the Universal Soil Loss Equation (USLE) was developed

in the 1930s to predict the long-term (roughly 22-years) annual average soil loss based on over 10,000 plot-years of

measurements (for a review see Alewell et al., (2019)). A large number of models have been since developed, ranging from

purely statistical to highly process-based, yet for numerous reasons none have surpassed the popularity of the (Revised)USLE

(Borrelli et al., 2021). Although the measurements used to develop the RUSLE were made at the event scale, the temporally

lumped predictive target of the (R)USLE is less prone to the effects of random variability compared to shorter temporal scales

(Nearing et al., 1999; Kim et al., 2016a). In turn, the legacy of the (R)USLE has meant a continued focus on the mean average

annual soil loss in the modelling community (Borrelli et al., 2021), with a comparatively much lower focus on the system

dynamics. Using the long-term annual average as the predictive target, focussing on the central tendency, arguably adds

simplicity when communicating soil erosion as a risk to farmers, policy makers and stakeholders, and removes the complex

temporal dimension when integrating soil erosion as a holistic component of soil health. Yet central tendencies poorly represent

the heavy-tailed nature of soil loss events, which add considerable uncertainty to long-term predictions by concealing the non-

linear internal dynamics (Kim et al., 2016b; Gonzalez-Hidalgo et al., 2009, 2012; Strohmeier et al., 2016).


The coincidence of heavy rainstorms with periods of high antecedent soil vulnerability (e.g. after a tillage event) embeds a

fundamental temporal dependency within soil loss (Nearing et al., 2005). For reasons of high stochastic variability, predicting

soil erosion and sediment delivery at dynamic timescales is a complex task (Batista et al., 2019; Kim and Ivanov, 2014; Kim

et al., 2016a), with signals that show strong scale-dependency (Walling, 1983; de Vente and Poesen, 2005; de Vente et al.,

2013, 2007; Gonzalez-Hidalgo et al., 2013). Nevertheless, understanding these phenomena at dynamic timescales can more

directly address how anthropogenic drivers, such as cropping system dynamics and soil management practices, interact with





the natural drivers of soil erosion (e.g. rainfall magnitude and intensity, soil properties, topographic gradient) to cause unsustainable soil loss (Eekhout et al., 2018). At the plot-scale, attempts to model the dynamics of soil erosion with simplistic models have well established their missing dynamic process components (Kinnell, 2010, 2005). For example, Kinnell, (2010)

outlines why deterministic runoff flux predictions through space and time are a critical uncertainty when applying the conventional (R)USLE model at the event-scale. Moving to larger spatial scales beyond the plot, the culmination of distributed uncertainty in overland flow predictions has limited their capacity to couple directly to erosion modules when predicting spatially distributed soil erosion rates (Jetten et al., 2003; Beven, 2006; Chaves and Nearing, 1991; Wendt et al., 1986).

Spatially distributed catchment models with many spatial elements need to find the balance between model error from unrepresented processes and spiralling error propagation from high parameter dimensionality (Van Rompaey and Govers, 2010). During the modelling process, sediment delivery predictions are most commonly calibrated and validated based on the spatially aggregated channel sediment yield record (Jetten et al., 2003). High model complexity is arguably poorly justified given the low relative quantity of information contained in a sediment yield time series, respective to plethora of hillslope and

valley-bottom processes (Jakeman and Hornberger, 1993; Govers, 2011). Experience from a range of modelling disciplines has outlines the conceptual and practical limitations of overcoming poor model performances by adding model complexity in the presence of poor data inputs (van Rompaey et al., 2010; Van Rompaey and Govers, 2010; Jetten et al., 2003; Batista et al., 2019). Low-complexity models have the further benefit of low parameter dimensionality which aid the human interpretation of model error (Sonneveld and Nearing, 2003; Paola and Leeder, 2011). Hence, there is justifiable value in testing the capacity

of simplistic models to model complex problems (Kinnell, 2010; Bagarello et al., 2017), or provide theoretical frameworks on which to build in the age of machine learning (Nearing et al., 2021). Modern information-dense geospatial information, such as Earth observation and other meteorological data systems, offer further unexplored potential to decrease the input parameter uncertainty by capturing the key spatiotemporal dynamics of rainfall and land-surface dynamics (Sidle et al., 2017; Refsgaard et al., 2022; Matthews et al., 2022, 2023a; Moller et al., 2014; Kim et al., 2020).


WaTEM/SEDEM (W/S) is the most widely applied spatially-distributed model to predict the long-term soil erosion and sediment delivery at the catchment scale in a spatially distributed manner (Alatorre et al., 2010; Van Oost et al., 2000; van Oost et al., 2002; Borrelli et al., 2018a, 2021). Its fundamental conceptions embed the RUSLE equation into a catchment-scale model by coupling it with a spatially-explicit sediment transport module to simulate erosion, deposition, and the portion of

sediment yield delivered to stream channels by rill and interrill erosion processes (Van Rompaey et al., 2001; Van Oost et al., 2000; van Oost et al., 2002). A key motivating factor for the uptake of W/S is the low number of free calibration parameters and an abundant reference literature (Sonneveld and Nearing, 2003; Borrelli et al., 2021; Alewell et al., 2019). Practicality has facilitated numerous spatially distributed catchment applications as well as continental-scale applications (Borrelli et al., 2018a), in this case allowing soil fluxes to be coupled with organic carbon. Numerous studies have established high temporal

variability in the dynamic (R)USLE factors from fine-scale observations, such as the rainfall frequency-magnitudes underlying



the R-factor (Bezak et al., 2021) and the seasonal dynamics of the C-factor in modern cropping systems (Gabriels et al., 2003). Yet, the long-term annual average focus of the (R)USLE model has meant a limited confrontation of these key dynamic drivers of gross erosion with catchment-stream sediment delivery at equivalent timescale.

In this study we investigate the capacity of several alternative W/S model structures to predict the multitemporal sediment yield. 'Multitemporal' is defined by one or more potentially erosive rainfall events per 15-day period, and the simulated sediment delivery is validated against instrumental measurements from the catchment outlets. A standardised parameterisation routine with dynamic crop cover and rainfall inputs is used in four catchments in North-West Europe to emphasise the importance of multi-catchment applications with inter-comparable posterior outputs. We firstly investigated: can W/S with a

temporally static calibration routine decompose the dynamics of sediment delivery from a multi-year aggregation of sediment yield? Secondly, does a multi-temporal calibration routine improve the model performance over a temporally static one? and can the seasonality of posterior output be used to infer the unrepresented processes responsible for the model error. Lastly, the implications of these two temporal approaches were further explored to investigate the interdependencies between space and time on the spatially distributed sediment delivery.


## 2 Catchment data subsamples from the EUSEDcollab database

Understanding the capacity of W/S to simulate the inter-annual variability of soil erosion-sediment delivery events requires catchment sediment yield measurements at a suitably small spatial scale (some suggest roughly $< 10$ km$^2$), ideally below the scale at which the signal of hillslope erosion is strongly confounded by other erosion (e.g. gullying, mass wasting, bank erosion)

or in-channel processes (de Vente and Poesen, 2005). Knowledge on the catchment erosion processes is likewise useful to understand if the dominant sediment sources match those simulated by W/S. We selected catchments meeting this criteria from the EUSEDcollab database (Matthews et al., 2023b), compiled with the specific purpose of providing publicly available catchment data suitable for soil erosion and sediment delivery modelling. EUSEDcollab contains measured time series data from monitored catchments across Europe, including measured discharge (Q), suspended sediment load (SSL), as well as

additional precipitation and land-use information where available.

Table. 1: Catchment overviews for the four monitored catchment sites simulated in this study. The catchments were selected for their relatively close proximity in Central Belgium and North-West France, meaning they are subject to similar climatic regimes as well as loess soil substrates. The metadata fields for each catchment correspond to the information available in

EUSEDcollab (Matthews et al., 2023b). The 'Data type in EUSEDcollab' refers to the temporal structure of the time series data as it is made publicly available. 'Event – variable timestep' signifies event data with a variable timestep within each event



due to the measurement method, while 'Event - aggregated' signifies event data with a singular aggregated sediment load per event. The 'Catchment name' field corresponds with the EUSEDcollab entry name.

| Catchment name | Country | Drainage area (ha) | Measurement extent | Data type in EUSEDcollab | Catchment literature |
|---|---|---|---|---|---|
| Ganspoel | Belgium | 117 | 1997/03/01 : 1999/03/01 | Event – variable timestep | (van Oost et al., 2005; Steegen and Govers, 2001; Steegen et al., 2001) |
| Kinderveld | Belgium | 250 | 1996/07/01 : 1999/11/01 | Event – variable timestep | (Steegen and Govers, 2001; Steegen et al., 2000, 2001; van Oost et al., 2005) |
| BRVL (Bourville) | France | 1045 | 2007/09/01 : 2018/08/31 | Event - aggregated | (Richet et al., 2020; Ouvry and et al., 2018; Grangeon et al., 2022; Pak and et al., 2018; Grangeon et al., 2020) |
| FDTL | France | 145 | 2011/11/01 : 2018/08/31 | Event - aggregated | (Richet et al., 2020; Ouvry and et al., 2018; Grangeon et al., 2022; Pak and et al., 2018; Grangeon et al., 2020) |


In this study, we utilised four catchment datasets with measurements taken in ephemeral stream channels activated during rainfall-runoff events (Fig. 1 & 2). The selected catchments, all situated on the European loess belt (Lehmkuhl et al., 2021), were as follows: Ganspoel, BE, Kinderveld, BE, Bourville (BRVL), FR and FDTL, FR, with a catchment drainage area of 117, 250, 1045 and 145 hectares (ha) respectively. The FDTL is a nested sub-catchment within the BRVL catchment, while

the Ganspoel and Kinderveld are in close proximity to each other (Fig. 1). The extreme event on May 20[th] 1997 in the Kinderveld catchment was not considered, since it was estimated in the study of (Steegen et al., 2000) and corresponds to a return period (ca. 10 years) considerably exceeding the measurement period. All catchments come with the advantage of being well studied over their measurement periods (Table. 1), providing a large depth of contextual information on the land management and erosion processes. In the case of the BRVL and FDTL, underlying karstic rock formations also influence the

infiltration dynamics of the catchments (Grangeon et al., 2022). Owing to loess-derived silt and loam soils, all catchments are susceptible infiltration-excess runoff and to winter soil surface crusting, particularly given the dominance of arable crop cultivations. The influence of soil crusting has been evidenced to drastically impact the infiltration in all catchments, typically developing through the winter to result in high runoff propensity during late-winter rainfall events until new tillage operations break the crusting surfaces (Grangeon et al., 2022; Steegen et al., 2000; Le Bissonnais et al., 2005).







**Figure 1. An overview of the catchment boundaries and gauge locations for the Ganspoel (BE), Kinderveld (BE), BRVL (FR) and FDTL (FR) catchments in North-West Europe. The point locations mark the sites of water discharge and suspended sediment sampling, while the polygons outline mark the delineated catchment boundaries. Base maps provided by ©Microsoft and ©ESRI.**





**Figure 2. Time series plots of the water discharge (blue) and suspended sediment load (red) per event for the Ganspoel (BE), Kinderveld (BE), BRVL (FR) and FDTL (FR) catchments. The total suspended sediment loads for four catchments are 193, 2390, 560 and 482 tonnes over their respective measurements lengths. The lower pie charts give the seasonal sum of the total suspended sediment load (SSL) per catchment.**



## 3 Methods

### 3.1 Hillslope soil erosion and sediment delivery modelling using WaTEM/SEDEM

A full explanation of the W/S model structure and functional equations is provided in its original literature (Van Rompaey et al., 2001; Van Oost et al., 2000; Verstraeten et al., 2006). The model links a cellular gross soil erosion quantity to a net soil loss delivered to stream channels by routing a sediment flux along the topographic gradient at a rate determined by the cell-wise transport capacity. Accordingly, gross erosion (A) is calculated in each cell via the RUSLE hybrid-statistical model:


$$A = R \, K \, LS_{2D} \, C \, P, \tag{1}$$

Where R is the rainfall erosivity per unit time (MJ mm ha$^{-1}$h$^{-1}$), K is the soil erodibility (t ha h ha$^{-1}$ MJ$^{-1}$ mm$^{-1}$), C is the crop cover and management factor (unitless), P is the physical interventions factor (unitless) and $LS_{2D}$ determines the

topographic factor (unitless) on a 2-dimensional grid (Desmet and Govers, 1996; Wischmeier and Smith, 1978).

In the standard model implementation, the transport capacity (TC) quantifying the maximum cell-to-cell flux rate across the face of each cell is approximated via:

$$TC_1 = ktc \, R \, K \, (LS_{2D} - 4.1 \, SIR), \tag{2}$$

By evaluating the total sediment mass (input flux + gross erosion) at every cell against the transport capacity, W/S also simulates the spatial patterns of net erosion and deposition. The inclusion of these two components allows the model to fluctuate between a detachment and transport-limited situation in different areas of the landscape. In intensively cultivates

landscapes the ktc parameter is differentiated between arable (ktc$_{high}$) and non-arable (ktc$_{low}$) land respectively, due to sharply increased erosion and sediment transport rates in the former. These two values are then assigned to the spatial grid and calibrated iteratively to the long-term sediment yield in multiple catchments.

All transport capacity formulae are influenced by the landscape structural components in W/S, which affect both the predicted

sediment delivery to varying degrees (Batista et al., 2022) and the spatial patterns of erosion and deposition in the landscape (Borrelli et al., 2018b). The parcel connectivity parameter defines how the total upstream contributing area is lost across a boundary transition to a new land-use type (cropland or forest). The parcel trapping efficiency scales the contribution of each cell to the total upstream area based on the cell's land-use type (cropland, forest or pasture). These parameters act to influence the spatial patterns of erosion and deposition based on the pattern of landscape heterogeneity.






### 3.2 Implementing WaTEM/SEDEM for multitemporal predictions

W/S was implemented in a 15-day temporal format to standardise the comparison between simulated and observed SSL (Table. 1). The standard temporal structure overcomes some of the issues involved in defining consistent thresholds of rainfall-runoff initiation when modelling discrete event episodes, which can impact the timing and total number of simulated events. Typically 195 W/S has a graphical user interface (GUI) framework, which although user-friendly, limits the opportunities to modify the model iteration routine and its interaction with other scientific software (Bezak et al., 2015). Moreover, in a dynamic modelling framework inclusion of time variant parameter layers vastly increases the computational demand compared to a temporally static routine. Therefore, W/S was implemented dynamically within an end-to-end parameterisation – implementation – validation Python workflow for increased efficiency when applied to multiple catchments (Fig. 3).


In this study, we tested two approaches to optimise the model prediction of net soil loss through time: 1) a temporally static calibration: each catchment is calibrated to obtain one single calibrated ktc parameter pair that converges the simulated sediment yield sum on the observed value, and 2) a multitemporal calibration framework: the calibrated ktc parameter pair is permitted to vary on a monthly basis to test how the model performance can improve if seasonal variability is added, and 205 moreover if useful inferences about the system can be made. In both cases, given that the catchment sediment yield time series records vary in their length and age, we don't attempt a strategy to find a global or stratified calibration parameter set, as would be typically common for W/S (e.g. de Vente et al., 2008). The temporally static calibration (1) is analogous to a temporal decomposition of the net sediment yield using the standard W/S calibration routine, with a structure which includes dynamic model parameters and a consideration of runoff in the TC equation. The multitemporal calibration framework (2) involves the 210 fitting of interpretable splines curves to describe the seasonality of ktc parameters, and thereafter to infer the necessary temporal variation in the missing process representations. We further discuss the opportunities and limitations of such an approach, and how it can be applied to inform future model development.





**Figure 3. A schematic overview of the standardised data pre-processing and model implementation routine for the selected European catchments. All stages are implemented within a python workflow using a common set of input data sources to promote inter-comparable modelling efforts between catchments. The full names for the schematic acronyms are: 1) The land use and land cover**



**survey (LUCAS), 2) The Integrated Administrative and Control System (IACS) field parcel data, and 3) Open Street Map (OSM) cartographical data.**

### 220 3.2.1 Adapting the sediment transport capacity for multitemporal modelling

Several recognised limitations exist when coupling gross erosion and sediment transport to the same topographic parameters, particularly in areas of intense flow convergence where incision can periodically occur and contribute to the sediment yield (Verstraeten et al., 2007). Henceforth, we investigate the potential of several alternative transport capacity formulae in W/S in addition to the standard implementation (Eq. 2). Firstly, the generalised stream power law:


$$TC_2 = ktc \, A^{1.4} S_T^{1.4},$$ (3)

Where area ($A$) is a temporally static surrogate for water discharge, $S_T$ is the topographic slope gradient, and the two exponent values of 1.4 are the median values reported across studies of experimental studies (see Prosser & Rustomji, (2000) for an in-
depth discussion). Secondly, a cell-wise runoff coefficient was introduced to the model to scale A based on the cellular propensity to produce runoff:

$$TC_3 = ktc \, (A \, R_C)^{1.4} S_T^{1.4},$$ (4)

Here, $R_C$ is a unitless coefficient defined by:

$$R_C = 1 + \frac{(\frac{R_{CN}}{RF} \times 100)}{D_E},$$ (5)

Where $R_{CN}$ is the runoff depth approximated by the curve number method (SI. 1.1), RF is the event rainfall depth (mm)
recorded by the catchment rain gauge, and $D_E$ is the rainfall event duration in days. For the purpose of creating a unitless proxy for runoff, $D_E$ is limited to 6 hours (0.25 days) to avoid cases in which $R_C$ becomes very high for events with short measurement duration periods, potentially due to highly localised precipitation in which only a portion is captured by the measuring gauge. Due to large differences in the magnitude of the ktc value for each TC formula (Verstraeten et al., 2007), the ktc parameter pairs for TC$_2$ and TC$_3$ were rescaled within the W/S software by dividing by $10^4$ and $10^6$ respectively.


Previous implementations of W/S over multiple time steps have noted issues with unrealistic cellular sediment deposition rates which exceed plausible patterns of landscape evolution (Van Loo et al., 2017). This is particularly true for TC formulae with exponents on the cellular drainage area and slope variables exceeding 1, which exaggerate errors in secondary attributes





derived from the DEM model surface. The potential reasons propagate from fundamental scale issues with limited process

representation, to instrumental data acquisition, to the various stages of algorithmic pre-processing (Wechsler, 2007). A cellular limit of 5 mm per 15-days was therefore set within the model code, meaning that outstanding sediment mass is continuously routed and deposited in a more diffusive manner.

### 3.2.2 Model parameterisation

Dynamic (15-day) and static input layers were processed multi- or single layers of input parameters respectively at a 25-m resolution. Rainfall timing and magnitude properties are the independent variable within the (R)USLE module (Kinnell, 2010), with established losses in accuracy when large-scale gridded meteorological forcings are used for its parameterisation (Bezak et al., 2022; Matthews et al., 2022). We consequently rely on the catchment records of measured rainfall in EUSEDcollab for these data inputs (Table. 1).


The input layers for all other model parameters were prepared using standard methods built upon data sources with at least an EU coverage (Fig. 3). An overview of all temporally static data layers is given in Table. (2) and a further explanation of the processing methods given in SI. (1.3). In the case of IACS field parcel data, full EU-coverage is in progress as the responsible agencies in EU Member States make the data publicly available. Efforts are in progress to harmonise and augment this data

for EU applications in multiple research domains (Schneider et al., 2023).

Table. 2: Geospatial data inputs used for the parameterisation of the temporally static parameter inputs. All inputs have a European coverage with the exception of IACS data which has an increasing coverage as the number of EU regions with open-access data increases.


| Parameter type | Data source | Type (resolution) | Reference |
|---|---|---|---|
| Field parcel declarations | IACS (multiple EU data regions) | Vector | (Schneider et al., 2023) |
| Topographic grid | EU-DEM | Raster (25-m) | (EU-DEM, 2023) |
| K-factor | ESDAC | Raster (500-m) | (Panagos et al., 2014) |
| Landcover | ESA WorldCover | Raster (10-m) | (Zanaga et al., 2022) |
| Roads and paths | OpenStreetMap API | Vector | (OpenStreetMap, 2023) |

### 3.2.3 Temporally dynamic RUSLE parameters

The RUSLE rainfall erosivity (R) and crop cover and management (C)-factors were treated as dynamic within the modelling framework. Rainfall erosivity was calculated per 15-days by integrating the individual EI30 events within the catchment





measurement gauge records (SI. 1.2). The annual C-factor is the rainfall erosivity-weighted sum of the soil loss ratio (SLR) in
each 15-day time period of the year:

$$C = \frac{(SLR_1 EI_1 + SLR_2 EI_2 + SLR_n EI_n)}{EI_t},$$ (6)

$$SLR = e^{FVC * \beta},$$ (7)

We use the decomposed C-factor (SLR) values per field parcel, in which each SLR represents the 15-day soil loss ratio
compared to that occurring on a bare fallow plot (Wischmeier and Smith, 1978). Here, the SLR was derived based on an
adaptation of the methods outlined in (Matthews et al., 2023a), but considering only green photosynthetic vegetation. This
method derives a continuous, gap-filled, 15-day time series of the SLR for each field parcel unit from the estimated total
fractional vegetation cover (FVC). A noteworthy consideration is that this method accounts for only the canopy-cover
component of the C-factor, which is the component receiving the overwhelming majority of research (Alewell et al., 2019).
Possible deviation in the temporal patterns of the SLR due to prior-land-use, surface-cover (i.e. crop residues), surface-
roughness and soil-moisture is therefore an outstanding source of model parametrisation error.

A key adaptation of the method was made in the Google Earth Engine workflow to replace the Sentinel-2 data with a
harmonised version of the Landsat 5, 7 and 8 collections to cover more extensive time periods. To mitigate against changing
field parcel structures through time, the 2018 IACS data layer was used and pixels were acquired from within a central polygon
around the 2018 parcel centroid (SI. 1.4). Additionally, crop-wise NDVI-canopy cover relationships were not considered due
to the historical nature of the catchment data compared to the IACS data (Table. 1). Due to limited acquisitions in some years,
parcels containing less than 10 Landsat pixel acquisitions per year were removed and replaced with the observations from the
closest year containing more than 10 acquisitions. While this involves parameterisation trade-offs, it allows more
representative land cover situations of the catchment measuring periods to be applied. The Pearson correlation coefficient (r =
0.66, p < 0.05) between the SLR and the count of bare fields in the Kinderveld surveys evidences a good correspondence
between the predicted and observed crop dynamics at the catchment scale (Fig. 4).



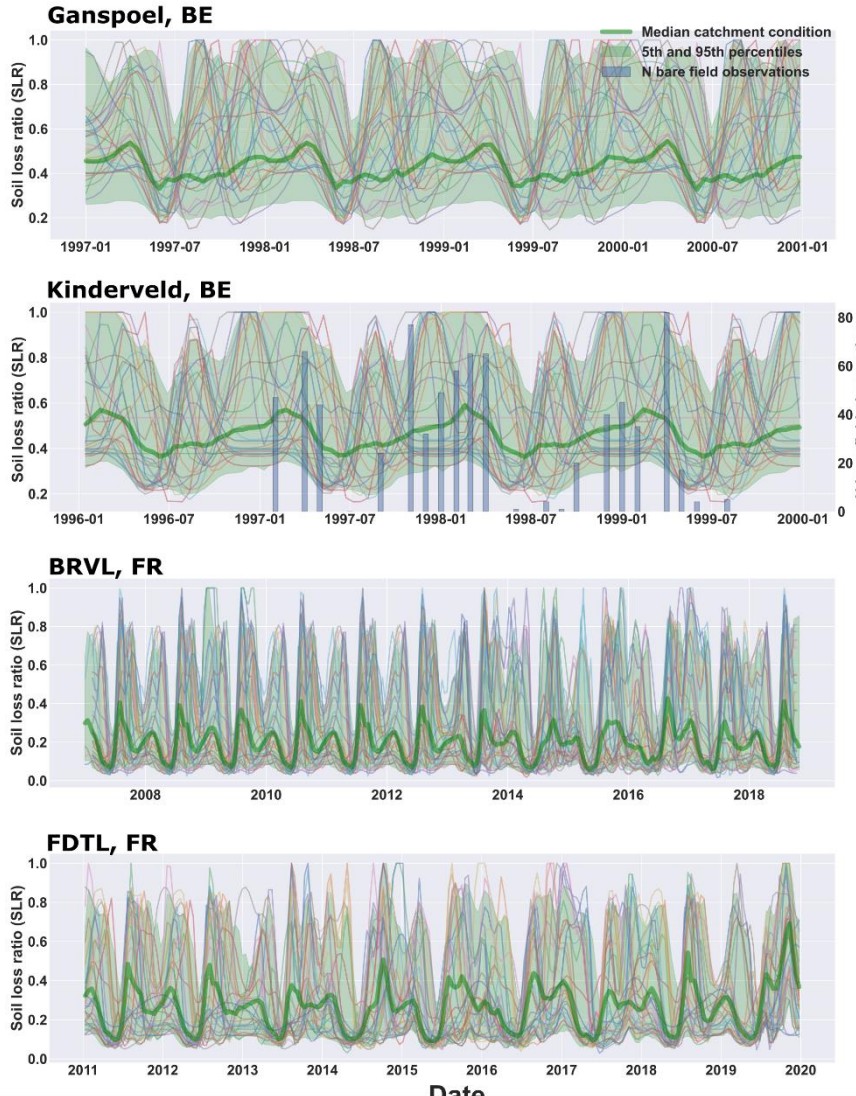

**Figure 4. Time series plots of the median SLR values from all arable field parcels within each catchment through time (green line) and the 5th and 95th percentile envelope. The multiple lines show individual examples of the time series from 20 field parcels within each catchment. Additionally, for the Kinderveld catchment, the 'N bare field observations' shows the bar chart count of bare fields within the catchment across multiple surveys.**



### 3.3 Model routine: temporal structure, calibration and validation

#### 3.3.1 Model evaluation based on the suspended sediment load

Within the standard model implementation, W/S is calibrated by varying the ktc parameter pair to fit the multi-year aggregate sum of net soil erosion measured in numerous instrumented stream channels or sediment trapping environments (e.g. reservoirs or sediment trapping ponds) (Van Rompaey et al., 2001). After calibration using a data portion, a spatial split is then made to test the model's ability to simulate the separate validation portion. Using an annual average with a sufficiently large event number permits more representative mean values (Gonzalez-Hidalgo et al., 2009), less influenced by the properties of the sample of erosion event magnitudes constituting the time series.

In all multitemporal implementations of W/S, the model was instead evaluated through time. The model performance was then evaluated using the Nash-Sutcliffe model efficiency (NSE) score:

$$NSE = \frac{\sum_{i=i}^{n}(y_{sim} - y_{obs})^2}{\sum_{i=i}^{n}(y_{obs} - \bar{Y}_{obs})^2} \;, \tag{8}$$

In which $y_{sim}$ is the simulated 15-day SSL, $y_{obs}$ is the observed 15-day SSL, and $\bar{Y}_{obs}$ is the observation mean. NSE ranges between one (a perfect model prediction) and -infinity, wherein a value below zero indicates that the mean gives a more informative prediction than the model (Nash and Sutcliffe, 1970).

#### 3.3.2 Temporally static model calibration

For each catchment time series, the multitemporal W/S simulation was repeated 100 times for different model connectivity scenarios and 15-day sediment delivery ratio (SDR) limits (SI. 1.5). By setting SDR-thresholds per scenario, we additionally investigate changes in model performances for the different transport capacity formula (eq. 2, 3 & 4) when events are included with more probable contributions from processes other than rill and interrill erosion. In each model scenario the 2 ktc parameters are iteratively changed to find the singular pair which minimises the difference between the aggregated sum of the simulated and observed SSL. The 15-day measurement records of SSL are then used to validate the ability of W/S to deterministically simulate, or decompose, the 15-day dynamics of the SSL. This draws upon the unique temporal information contained in records from small instrumented catchments but is absent from sediment trapping environments (e.g. small reservoirs and trapping ponds). However, using this abundant calibration target the model implementation could practically extend to more abundant long-term records of sediment yield (Vanmaercke et al., 2011).



### 3.3.3 Temporally dynamic model calibration

By limiting the calibration procedure to only the ktc parameter pair, one can plausibly investigate if residual seasonal dynamics
are present through employing a multitemporal calibration. In this case, ktc through time is a surrogate parameter for time-varying phenomena which affect the transport of sediment in the catchment (e.g. vegetation development, soil moisture condition, root cohesion, soil surface crusting etc). To avoid this optimisation routine being a black box, a posterior interpretation is made to relate the optimised ktc to the dynamic input parameters (Section 3.2.3) as well as documented phenomena within each catchment from published studies (Table. 1).


Using the best performing model connectivity parameters from the best model run from the standard area-slope transport capacity ($TC_2$) in W/S (Section 3.3.2), we define an optimisation problem with 6 parameters. The first 5 parameters fit the most likely $ktc_{high}$ curve as a function of the month of the year, which is calculated as the weighted sum of five B-splines basis functions of order two, defined recursively as described by Lancaster and Šalkauskas, (1986). The sixth parameter is the
$ktc_{low}/ktc_{high}$ ratio, which determines the monthly $ktc_{low}$ value as a fraction of $ktc_{high}$, enforcing an ordering relationship between the two. W/S was then run iteratively within the scipy.optimize.minimize package, which efficiently modifies the 6 aforementioned parameters using the Nelder-Mead method until an objective function describing the similarity between the prediction and observation is minimised (Gao and Han, 2012; Nelder and Mead, 1965). In this case, the objective function adopted was the root mean square error (RMSE) of the 12 simulated versus observed mean monthly sediment yield. We ruled
out the possibility of using the model for prediction in unknown locations or time periods given that the calibration time series is not completely distinct from the data used in the validation procedure (Section. 3.3.1). Furthermore, no extra procedures were adopted to prevent overfitting, such as cross-validation or penalisation of the objective function.

## 4 Results

### 4.1 Evaluating a temporally static model calibration over multiple scenarios

A temporally static calibration exhibited a poor predictive skill when applied to decompose the 15-day dynamics suspended sediment load from the temporally aggregated sum. In Table. (3) the best model runs for the different run scenarios are given per TC formula. A general ranking of model performance between catchments was observed: Ganspoel (NSE = 0.28) > Kinderveld (NSE = - 0.09) > FDTL (NSE = - 0.04) > BRVL (NSE = - 0.03). Despite a generally poor predictive skill, $TC_2$ implemented in W/S evidenced the best performance from a singular run (Table. 3), and showed a lower rate of model skill
loss at higher SDR values (SI. 2). Introducing a cell-wise runoff coefficient ($TC_3$) reduced the average model efficiency relative to its equivalent based on topographic factors only ($TC_2$), and increased the model sensitivity to changes in connectivity sub-parameters (SI. 2). The ranking of model performance between catchments corresponded reasonably well to the correlation between the suspended sediment load and the mean predicted RUSLE gross erosion: Ganspoel (r = 0.64, p < 0.05) > Kinderveld



(r = 0.23, p > 0.05) > FDTL (r = 0.18, p > 0.05) > BRVL (r = 0.04, p > 0.05). Given the relatively similar monthly distributions
of gross erosion between catchments, this resulted in a relatively higher model performance in the catchment without high
winter sediment loads (Fig. 2 & 5).

Table 3: A summary of the best catchment model runs per transport capacity model (TC model) from the alternative model
spatial structure scenarios. The transport capacity model with the relatively highest Nash-Sutcliffe model efficiency (NSE)
score is bold and underlined per catchment. Plots showing the model performance from all simulations can be found in Fig.
(S1). Note that both ktc pairs for $TC_2$ and $TC_3$ are rescaled within the W/S software by dividing by $10^4$ and $10^6$ respectively.

| Catchment | TC model | $ktc_{low}$ | $ktc_{high}$ | Simulated SSL sum (% of observation) | NSE | N periods (15-day) | SDR (max) | Connectivity index |
|---|---|---|---|---|---|---|---|---|
| Ganspoel | **$\underline{TC_1}$** | 4.06 | 8.13 | 95 | 0.284 | 18 | 0.235 | 2.0 |
| Ganspoel | $TC_2$ | 1.30 | 2.59 | 118 | 0.187 | 18 | 0.235 | 2.6 |
| Ganspoel | $TC_3$ | 2.21 | 4.41 | 89 | -0.072 | 18 | 0.235 | 1.6 |
| Kinderveld | $TC_1$ | 18.25 | 52.25 | 102 | -1.556 | 31 | 0.388 | 3.9 |
| Kinderveld | **$\underline{TC_2}$** | 9.13 | 18.25 | 89 | -0.091 | 31 | 0.388 | 2.9 |
| Kinderveld | $TC_3$ | 22.97 | 45.94 | 77 | -1.870 | 31 | 0.388 | 4.5 |
| BRVL | $TC_1$ | 5.00 | 10.00 | 69 | -0.080 | 52 | 0.027 | 0.8 |
| BRVL | **$\underline{TC_2}$** | 0.37 | 0.74 | 90 | -0.035 | 52 | 0.027 | 2.8 |
| BRVL | $TC_3$ | 5.00 | 10.00 | 53 | -0.189 | 52 | 0.027 | 2.0 |
| FDTL | $TC_1$ | 2.28 | 6.53 | 91 | -0.563 | 34 | 0.154 | 3.4 |
| FDTL | **$\underline{TC_2}$** | 2.07 | 4.14 | 82 | -0.043 | 34 | 0.154 | 2.5 |
| FDTL | $TC_3$ | 25.00 | 50.00 | 37 | -0.591 | 34 | 0.154 | 1.3 |





**Figure 5. Overlain plots with boxplot distributions of the 15-day RUSLE gross erosion per grid cell with bar plots of the total monthly sum of the suspended sediment load (SSL). A relative divergence represents months in which the predicted dynamics of gross erosion are more decoupled from the catchment SSL. Missing months are those without measured sediment yield in the 15-day time series.**

## 4.2 Multi-temporal model calibration

Implementing a multi-temporal calibration scheme using $TC_2$ significantly improved the model performance in all catchments (Fig. 6 & Table. 4). An interpretation of the model residual errors shows that the temporally static version of W/S results in a



systematic overprediction of the 15-day sediment load at low values by accounting for seasonal variability in the transport capacity (Fig. 6). The temporally dynamic calibration routine corrected for this systematic error when boosting the model performance. In doing so, the high temporal variability in the derived ktc surface signified that important residual error is 385 unaccounted for when using a temporally static transport capacity.

Table 4: Per catchment model performances from introducing a temporally dynamic model structure into W/S using $TC_2$ (eq. 3). The evaluations are made for the 15-day time series of simulated and observed suspended sediment load over the complete catchment measurement period. Note that the ktc pair for $TC_2$ are rescaled within the W/S software by dividing by $10^4$.


| Catchment | NSE | Simulated SSL sum (% of observation) | Mean monthly ktc value | Min monthly ktc value | Max monthly ktc value |
|---|---|---|---|---|---|
| Ganspoel | 0.42 | 94 | 2.78 | 0.02 | 10.35 |
| Kinderveld | 0.48 | 99 | 17.23 | 3.08 | 45.41 |
| BRVL | 0.63 | 102 | 2.62 | 0.1 | 18.04 |
| FDTL | 0.39 | 95 | 0.74 | 0.04 | 1.69 |



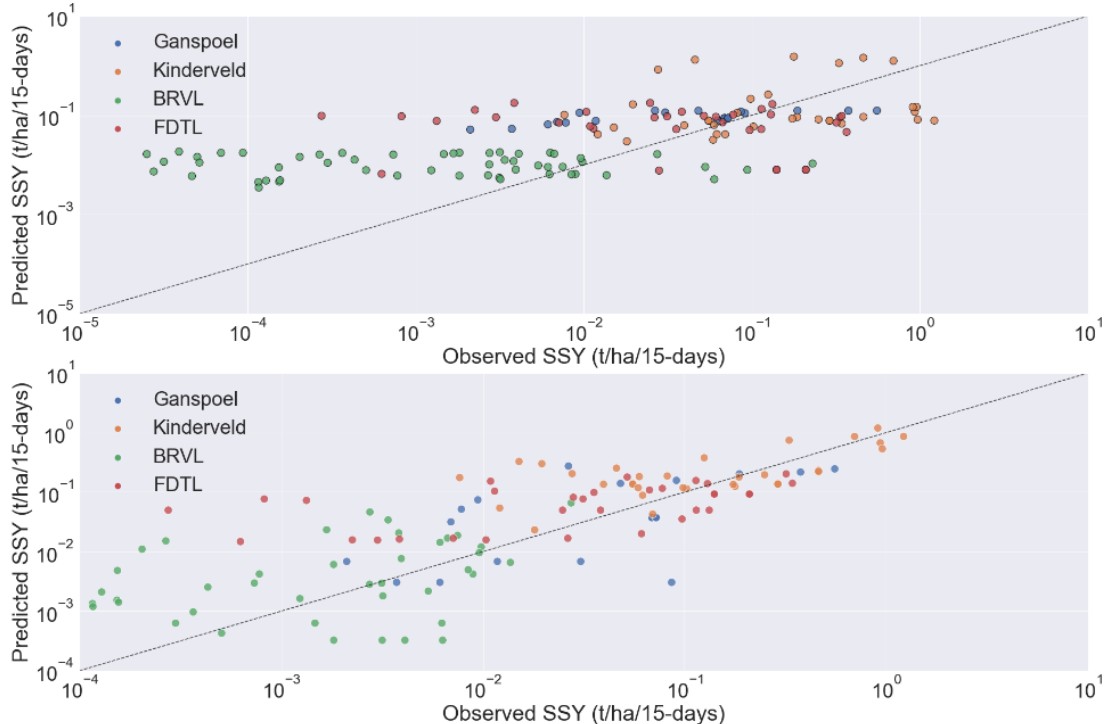

**Figure 6. Comparisons of the simulated and observed area specific sediment yield compiled for all catchments. Upper: the temporally static W/S calibration routine with a single parameter set. Lower: the temporally dynamic W/S calibration routine allowing monthly variation in the ktc parameters.**

The posterior monthly patterns of the optimised ktc parameters displayed seasonality which corrects error in the represented model processes and parameter inputs. For the Kinderveld, BRVL and FDTL catchments, the posterior interpretation of the ktc temporal curves showed significant reduction in the ktc parameters in summer months relative to the winter months (Fig. 7). These examples exhibit cases of loosely-coupled gross erosion and sediment delivery, the correlation coefficients between the 15-day mean predicted gross erosion and measured suspended sediment load were 0.23, 0.18 and 0.04, respectively. As mentioned, the Ganspoel catchment represented an exceptional case in which the predicted gross erosion and suspended sediment load were more tightly coupled. In all catchments except the Ganspoel, temporal decoupling was signified by a negative correlation between the predicted RUSLE gross erosion and the SSL (Table. 5). Time periods with a high ktc, typically occur in winter, in which a tighter coupling is necessary between the limited quantity of predicted gross erosion and the sediment delivered to the catchment outlet (prevailing detachment limitation). In contrast, periods with a low ktc typically





prevail in late-summer in which high quantities of gross erosion have a limited transport capacity (prevailing transport limitation).

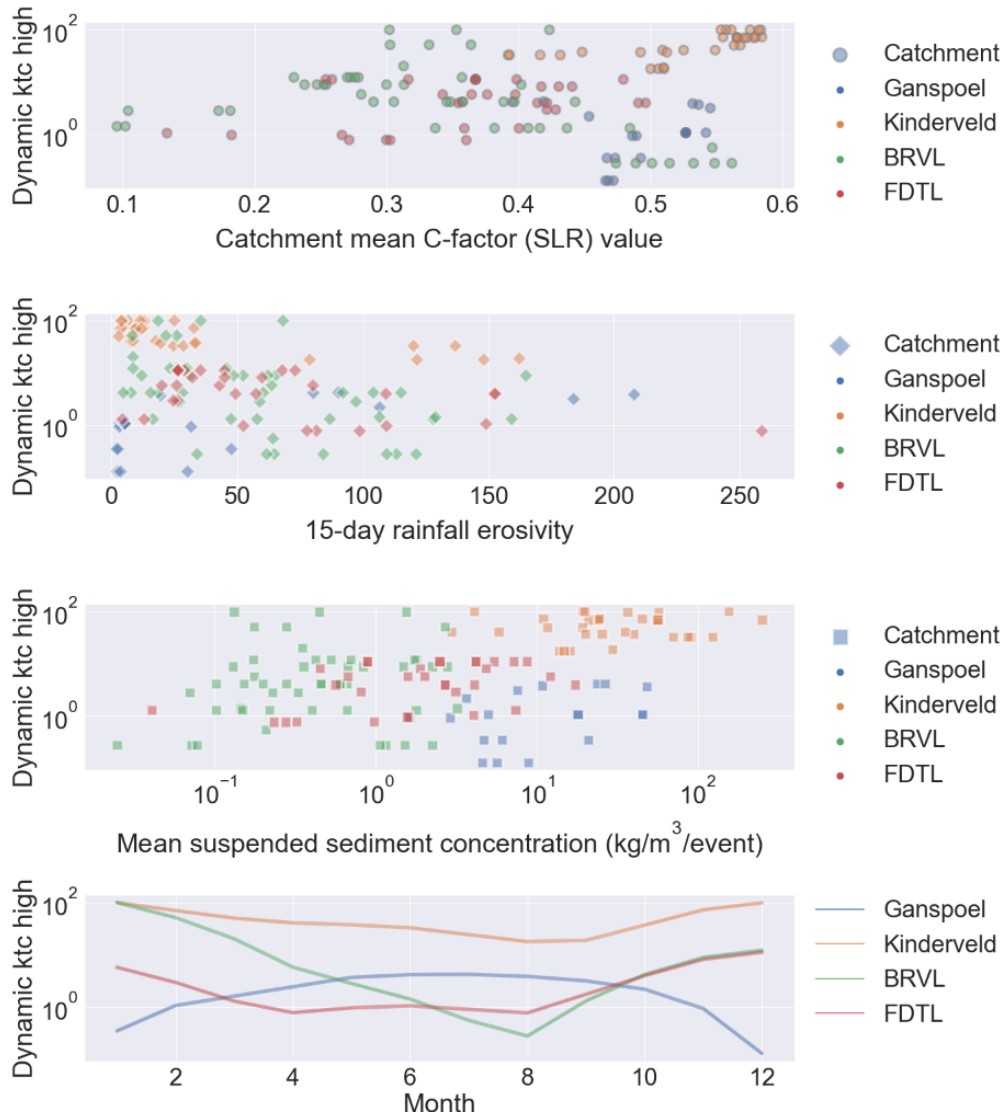


**Figure 7. Multiple plots giving a posterior overview of the monthly variable ktc$_{high}$ variable per catchment, as a function of: 1) the 15-day soil loss ratio (SLR), i.e. the temporally dynamic component of the C-factor, 2) the 15-day rainfall erosivity value, 3) the measured suspended sediment concentration per event at the catchment outlet, and 4) the plotted monthly ktc value resulting from the multi-temporal calibration showing the relative seasonality of the transport capacity.**


When all catchments were combined, a negative correlation is found between ktc and rainfall erosivity through time ( r = - 0.344, p < 0.05), signifying that high rainfall erosivity values need generally to coincide with lower landscape-scale transport





capacity (ktc). Contrastingly, higher catchment SLR values, which occur more commonly during winter periods with low

vegetation cover, correspond with higher ktc values through time ($r = 0.402$, $p < 0.05$). However the relationships between the

fitted ktc and the dynamic input parameters or measured SSL does not generalise in all circumstances (Table. 5). In the case

of the Kinderveld catchment, evaluating the dynamic ktc value against the mean catchment properties from surveyed fields

revealed both a negative Spearman's rank correlation with the observed vegetation cover ($SR = -0.624$, $p < 0.05$) and a positive

correlation with the soil crusting state ($SR = 0.365$, $p < 0.05$); indicative of higher ktc values during periods with low crop

cover and soil crust developments (Fig. 8).


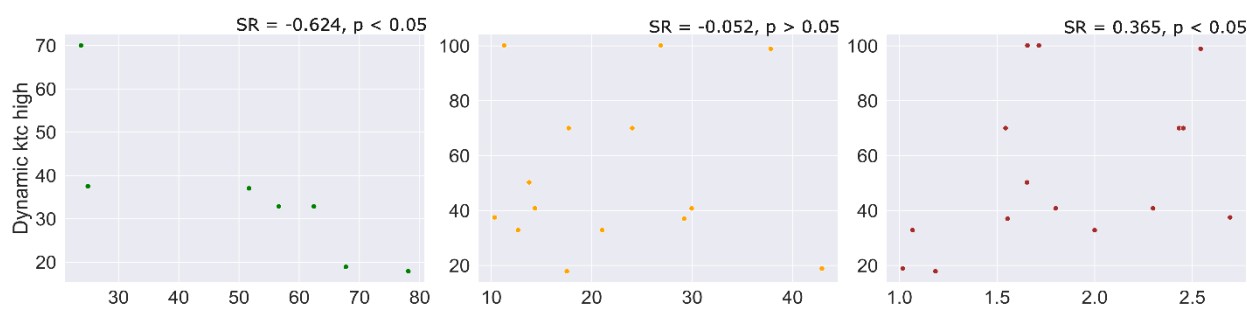

**Figure 8. Scatter plot relationships between the monthly survey values of spatially distributed field cover properties and the monthly ktc$_{high}$ value resulting from the multi-temporal calibration. The ktc$_{high}$ value assigned to each 15-day period is accordingly matched with the closest survey observation of the catchment. Only survey events exceeding 50 observed fields are included within the dataset.**
**A full explanation of the catchment survey procedure is given in** (van Oost et al., 2005)**.**

Table 5: A multiple Pearson correlation coefficient (r) matrix between the derived ktc value per month and multiple input

parameters or measured features of the suspended sediment load at the catchment outlet. Underlined Pearson r values are

significant beyond the $p < 0.05$ limit.

| Catchment | ktc vs RE | ktc vs SLR (catchment mean) | ktc vs gross erosion (mean) | ktc vs SSC (kg m-3) | ktc vs SSL (t ha-1 event-1) |
|-----------|-----------|------------------------------|------------------------------|----------------------|------------------------------|
| Ganspoel | 0.701 | -0.117 | 0.611 | 0.337 | 0.607 |
| Kinderveld | -0.644 | 0.684 | -0.653 | 0.109 | -0.001 |
| BRVL | -0.281 | -0.053 | -0.206 | 0.044 | 0.362 |
| FDTL | -0.414 | 0.22 | -0.304 | 0.174 | 0.478 |






Figure. (8a) shows the time-aggregated spatial patterns of the erosion and deposition estimates for the Kinderveld catchment, produced using the multitemporal calibration method. Within the dynamic model version, transitionary areas of the catchment can switch between both erosion and deposition through time, depending on the event conditions (Fig. 9b). For the Kinderveld catchment, the spatial outputs of W/S visually show a moderate capacity to identify areas of the catchment with surveyed

erosion and deposition features during the measurement period (Fig. 9c & d). Permitting temporal variation in the ktc parameters had a resulting impact on the landscape-scale fluxes in W/S (Fig. 9e). In catchments with negative correlations between the predicted gross erosion and the ktc values (Table. 5), landscape-scale transport limitation is more likely to coincide with events producing high amounts of gross erosion. Based on the multitemporal ktc curve for the Kinderveld catchment, the dynamic ktc values showed a strong increase in winter and a reduction in summer (Fig. 7). The model interlinkages between

the temporal dynamics and resultant spatial patterns show that a temporally static calibration of W/S results in relatively higher fluxes on hillslopes, compensated by higher deposition rates in thalwegs (Fig. 9e).



**Figure 9. Spatial outputs of WS for the Kinderveld catchment, BE containing catchment survey data** (van Oost et al., 2005)**. The individual panels show spatial outputs from W/S showing: a) the net sum of erosion/deposition over all erosive 15-day periods (n = 31) for the Kinderveld, BE, b) the binary count of events in which each pixel experienced erosion (-1) or deposition (+1) over the 31 simulated 15-day periods, c) a comparison of areas with a net deposition >= 0.5 mm with surveyed depositional features over the entire study period, d) a comparison of areas with a net erosion <= -0.5 mm with surveyed rill (yellow) and gully (red) features over the entire study period, and e) a comparison of the spatially distributed magnitudes of erosion and deposition for the temporally static and dynamic model runs. Red (green) areas indicate cells in which the predicted depth of erosion was higher (lower) in the static model version. Likewise, blue (yellow) areas indicate cells in which the deposition was higher (lower) in the static model version. Masked cells indicate the model stream channel position and the small number of cells in which the static and dynamic model versions disagreed between deposition and erosion. Base maps provided by ©Microsoft.**



## 5 Discussion

### 5.1 On the general applicability of W/S for dynamic modelling of gross and net erosion

Considerable simulation error is perhaps predictable when transposing models designed for time-aggregated temporal scales to describe the variability resulting from a complex suite of processes with complex spatiotemporal patterns (Kim et al., 2016b; Zhang et al., 2013). Although low-complexity models lack the full range of process considerations for temporally-dynamic predictions, there is both evidenced potential and a strong case for model testing and iterative development which avoids high parameter demand (Kinnell, 2010). Some key syntheses from the outlined model temporal downscaling attempt include: 1)

W/S performs comparatively worse in cases in which the multitemporal RUSLE predictions of gross erosion poorly correlate with the catchment SSL, 2) decoupling the gross erosion and sediment delivery modules to varying degrees through the year can correct for error in the temporal dynamics, which emphesises potential parameterisation error in the RUSLE predictions or missing model processes at dynamic timescales, and 3) the confounding influence of omitted time-dependent processes impacting runoff (e.g. antecedent soil conditions, overland flow re-infiltration, flow connectivity pathways, inter-event

deposition) as well as unconsidered erosion processes (e.g. gullying and bank erosion) may induce highly non-linear coupling between gross and net erosion at dynamic timescales. Furthermore, large differences in the obtained calibration coefficients from the different model applications reflect the challenges when seeking a 'global' calibration regime with sediment yield records of varying length (2 to 11 years) in small catchments with strong magnitude differences in sediment yield (Gonzalez-Hidalgo et al., 2013).


Evidence from the four modelled catchments corresponds to established knowledge that SSL is often non-linearly related to gross erosion (Sidle et al., 2017; de Vente et al., 2013; Parsons et al., 2006; Cerdan et al., 2004). In three of the four modelled catchments, the predicted temporal signal of gross erosion from the RUSLE module did not strongly correlate with the measured SSL. In these cases a temporally static sediment delivery module strongly impeded the model performance (Fig. 5).

Winter runoff events dominate the runoff and SSL budget in the BRVL and FDTL catchments (Grangeon et al., 2022), representing cases in which the seasonal dynamics of predicted gross erosion and measured SSL were inverted. Contrastingly, in the Ganspoel catchment, the predicted dynamics of gross erosion explained a considerable amount of the variability in the sediment yield ($R^2 = 0.41$), therefore resulting in a relatively higher model performance (NSE = 0.28). The occurrence of one high correlation suggests that a globally low accuracy in the parameterised dynamic variables is not the case, although this

cannot rule out changing parameter input accuracies or unconsidered processes impacting the gross erosion dynamics in different catchments (Kinnell, 2015). For example, studies in experimental catchments highlight that the parametrisation accuracy of rainfall-derived variables will decrease quickly as catchment size increases, due to fine-scale variability in erosive rainfall cells (Fiener et al., 2019).



Decomposing error in the multitemporal signals of the predicted gross erosion from the landscape-scale delivery of sediment
is fundamentally difficult due to the limited information content within the in-stream SSL (Govers, 2011). In heterogeneous
landscapes, decoupling between gross and net erosion magnitudes can occur through abundant sediment detachment during
summer storms corresponding with widespread transport limitation due to re-infiltration and sediment trapping at vegetative
barriers (Beuselinck et al., 2000). The spatial proximity of field parcels undergoing erosion to channels is therefore a key

feature in determining the outlet response (Steegen et al., 2000). In addition to important spatial components of the landscape
(parcel configuration, path and drainage channel distributions, topographic position), a sediment delivery module capable of
accurately simulating the multitemporal SSL requires a consideration of the processes producing and hindering runoff (Kinnell
and Risse, 1998; Kinnell, 2010). Runoff generation and throughput from upslope areas also remain the key determinant of
sediment export in three-dimensional landscapes at spatial scales beyond the plot (Kinnell, 2007; Takken et al., 2001).

Mirroring this study at the catchment scale, the systematic transition in W/S model error from overprediction to underprediction
with increasing event magnitude is also evidenced at the plot scale (Kinnell, 2010; Tiwari et al., 2000), resulting from a poor
consideration of runoff in the traditional RUSLE parameterisation (Foster et al., 1982).

In this study, the addition of a pixel-based CN runoff coefficient to the transport capacity equation (TC$_3$) was made through a

simplistic empirical model which matches the low parameter demand of the (R)USLE. The consequential decrease in model
performance occurred for two likely reasons: 1) the introduction of a high cell-wise sensitivity within the sediment routing
module (SI. 2), and 2) incorrect temporal profiles of predicted runoff which can cause their divergence from the RUSLE
parameters through time (Fig. 10). Both reasonings link back to the potential cascading error when explicitly including
overland flow, in that accurately describing its spatial and temporal magnitudes and threshold behavior introduces high model

sensitivity, with incorrect predictions hindering the model performance (Gómez et al., 2001; Vigiak et al., 2006; Merritt et al.,
2003).

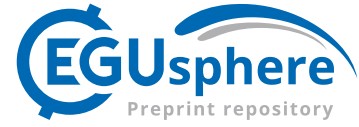



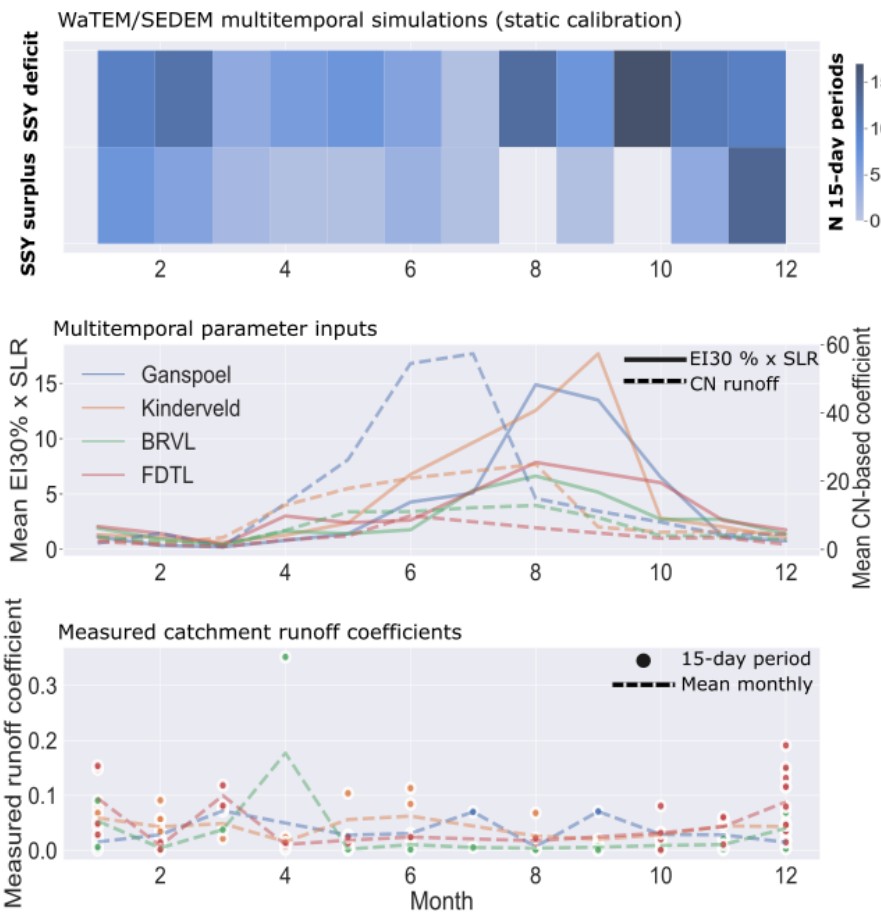

**Figure 10. Overview of the temporal divergence between the model temporal performance, input parameters, and the actual measured situation. Top: The monthly count of 15-day simulations that were underpredicted (SSY deficit) and overpredicted (SSY surplus) by the W/S model with a temporally static calibration. Middle: The mean monthly dynamics per catchment of: 1) the EI30 value multiplied by the soil loss ratio (SLR), the two key dynamic RUSLE parameters determining the seasonal dynamics of gross erosion, and 2) The CN runoff coefficient used to parameterise the dynamic transport capacity $TC_3$. Bottom: The measured mean monthly runoff coefficients (dashed lines) and individual 15-day values (points with corresponding colours for each catchment) which highlight the high runoff coefficients possible during winter months.**

High gross erosion rates are predicted between early summer and autumn using the dynamic RUSLE (Fig. 10), during which high intensity rainstorms coincide with seedbed preparation for summer and winter crops respectively (Vandaele and Poesen, 1995). For the Belgian Loess Belt (the Kinderveld and Ganspoel), this time period is also more susceptible to major muddy flood events (Verstraeten and Poesen, 1999). Nevertheless, relating these gross erosion dynamics to the catchment-scale sediment delivery may require an expansive consideration of hydrogeomorphic processes promoting erosion and deposition (Sidle et al., 2017). For example, the USLE-M model and its lineage of revisions follow the numerous parameters which are required to vary in order to adequately reflect the event-scale erosion at the plot scale (Kinnell and Risse, 1998; Bagarello et




al., 2017). The dynamic RUSLE parameter components (EI30 x SLR index) may exaggerate the disparity between summer and winter erosion rates (Fig. 10), principally due to the unconsidered antecedent soil conditions which impact erosion. A

winter erosion risk is particularly prevalent in management situations leaving loamy soils absent of winter cover, allowing the development of surface crusting followed by enhanced runoff and the development of rill and gully networks (Vandaele and Poesen, 1995; Le Bissonnais et al., 2005). These divergences between the temporal patterns of predicted gross erosion and the surveyed situation may attribute to several causes: 1) the EI30 sub-units of the R-factor, being derived to predict rill and interrill erosion, inflates the impact of infiltration-excess overland flow over saturation-excess overland flow at spatial scales beyond

the unit-plot; an explanation for relatively higher ratios of gully to rill ratios during winter in Flanders (Vandaele and Poesen, 1995), 2) missing antecedent factors causing changes in the soil erodibility (i.e. through the RUSLE K-factor) and landscape-scale water and sediment conveyance with developing soil moisture conditions, soil management practices, and the development of sealed soil surfaces (Steegen et al., 2000), or 3) underestimations of the secondary impacts of high vegetation cover on erosion and sediment delivery, such as the creation of seasonal vegetative trapping barriers which decrease both the

on-site gross erosion (i.e. through the RUSLE L-factor) and prevent its downslope transport .

## 5.2 The coupling and decoupling of gross and net erosion through time

### 5.2.1 Seasonal differences in sediment transport efficiency

Considering that optimisation can lead to uncertain interpretations in the case of multiple acceptable parameter combinations

(Beven, 2006; Brazier et al., 2000), inferences should be confronted with the available catchment observations and knowledge (Table. 1). The characteristics of sediment delivery are multifaceted and involve a high number of static and dynamic information components (Sidle et al., 2017), emphasising the importance of spatial surveys when seeking to represent spatial processes (van Oost et al., 2005; Steegen et al., 2001). Within W/S, the ktc parameter represents the propensity of the landscape to produce runoff and convey eroded sediment based on the properties of rainfall intensity, soil characteristics (e.g. erodibility,

roughness, infiltration rate) and vegetation characteristics (Verstraeten et al., 2007). Within a multitemporal calibration, we remove the assumption of temporal stationarity and instead define an optimisation routine to derive their most likely temporal profile, serving as a proxy for missing parameter information or process components (Fig. 8). Hypothetically, the multitemporal optimisation routine would derive a flat ktc curve, equivalent to the temporally static calibration routine, in a situation in which all variability in the response variable (15-day sediment load) were accurately described by the model.


The sediment delivery regimes in three modelled catchments were weakly coupled to the gross predicted erosion in summer months, signified by a negative correlation between EI30 and the calibrated ktc value. In all catchment situations, EI30 is strongly influenced by intense summer storms in North-West Europe which tend to dominate the long-term budget of rainfall erosivity (Ballabio et al., 2017). In contrast, the efficiency of sediment conveyance needs to increase in winter months, during



which the seasonal observations of runoff decouple most strongly from the RUSLE predictions (Fig. 9). Multitemporal catchment observations show that winter soil crust development on bare soils is critical in producing highly elevated runoff coefficients (Grangeon et al., 2022). Similar observations regarding crust development were made in the Kinderveld catchment, in addition to the heterogeneous 'filtering' effect of high summer vegetation growth conditions, which increased winter suspended sediment concentrations respective to those in summer (Steegen et al., 2000). At present, these time variances in antecedent soil conditions and transport capacity losses along the hillslope arguably represent key processes required to improve the deterministic model performance using a temporally static calibration.

The shape of the derived ktc curve through time from the temporally dynamic model calibration routine can be interpreted to represent differences in the structural and functional components of sediment connectivity (Heckmann et al., 2018). Structural and functional represent the spatial configuration of the landscape components and the actual dynamics of water and sediment transfer respectively (Wainwright et al., 2011). For two catchments in close spatial proximity, the Kinderveld is an example of a catchment with high structural connectivity compared to the Ganspoel (for an explanation see: Steegen & Govers, (2001)). The observed inverse relationship in the seasonal dynamics of the ktc profile between the Ganspoel and Kinderveld catchments suggests that the structural connectivity may strongly influence the functional connectivity, to an extent in which their temporal dynamics are highly different.

### 5.2.2 The spatial implications of a multitemporal modelling approach to sediment delivery

An array of erosion processes (e.g. mass wasting, gully erosion, bank erosion) contribute significantly to the sediment yield (Vanmaercke et al., 2011), particularly at larger spatial scales. In these cases, the restricted process considerations of W/S result in a recognised spatial misattribution of soil erosion source areas if the proportion of sediment yield isn't properly apportioned to the correct erosion processes (de Vente et al., 2013). This misattribution can be improved to some extent by model structural changes allowing high flow convergence within the transport capacity formula (Verstraeten et al., 2007). Here we further demonstrate how temporal error, in addition to process error, may propagate through the model design and potentially produce differences in the long-term spatial fluxes (Fig. 9). For this reason, the dynamic and temporally static W/S implementations cannot be considered as independent entities when determining the magnitudes of spatial fluxes. The temporally static and dynamic calibrations of W/S mostly correspond on the areas experiencing erosion and deposition, highlighting the importance of model structural differences in improving spatial accuracies (Batista et al., 2022; Vigiak et al., 2006). Nevertheless, non-linear temporal differences in ktc back-propagate over the landscape to change the magnitudes of erosion and deposition (Fig. 9). These findings, building on similar findings for gross erosion (Borrelli et al., 2018c), highlight the importance of improved representations of underlying temporal dynamics when seeking to upscale model predictions in space and/or time.





### 5.3 Future opportunities through model development, standardised parameterisation and data resources

Multi-catchment studies exploring the multitemporal properties of soil erosion and sediment yield are significantly more rare than single catchment applications (Fig. 11), which has arguably hindered understanding on how models can generalise across environments. Mirroring sister research domains (Nearing et al., 2021), process generalisation and upscaling will remain a

future challenge, ideally aided and not superseded by data driven models, particularly as data availability increases and multi-catchment applications can expose current model limitations (Matthews et al., 2023b). Interpretable statistical methods, such as multi-temporal curve fitting for optimisation, can reveal new opportunities to find the balance between improving the predictive performance through 'brute force' methods and adding model complexity to describe the dynamic processes (Refsgaard et al., 2022). Nevertheless, useful inferences can be made by an end-user by interpretable statistical model

components which may have the same desired effect as a fully process-based model. Here we show the opportunity for a temporally dynamic optimisation scheme to act as a proxy to reveal missing processes, which can in turn aid the iterative improvement of simplistic models.

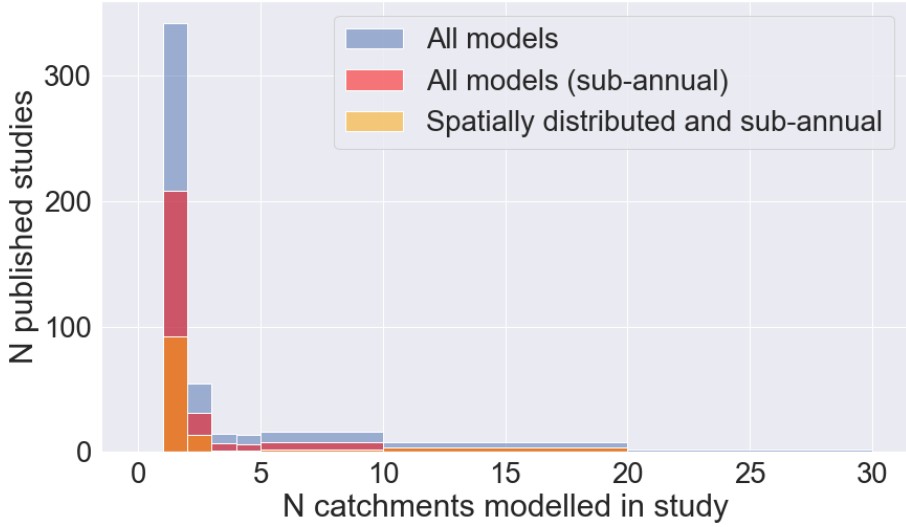

**Figure 11. An analysis of the Global Applications of Soil Erosion Modelling Tracker (GASEMT) database (Borrelli et al., 2021) showing the number of catchments modelled per study with a catchment scale focus. The dataset was subset into 3 types, to give an overview of the count of catchments per study: 1) 'All models' considers all GASEMT model entries at the catchment scale irrespective of their spatial structure (spatially distributed or (semi-)lumped, 2) 'All models (sub-annual)' includes all model spatial structures but with a sub-annual temporal resolution, 3) 'Spatially distributed and sub-annual' includes only grid-based model**
**applications at sub-annual temporal resolutions.**

The spatial characteristics of soil erosion represent the source of the cascading environmental impacts, arguably making them the key prediction target (Vigiak et al., 2006; Jetten et al., 2003; Merritt et al., 2003). While error on the spatial patterns of soil erosion and sediment transport can confound within the spatially lumped sediment yield, the spatial patterns can remain poor



(Jetten et al., 2003). Poor spatial performance of soil erosion models to perform binary classification tasks has been previously noted (Evans and Brazier, 2005), and arguably presents a key challenge when linking models to parcel-scale soil erosion risk. Within standardised model parameterisation workflows, abundant information streams can potentially improve model performances by describing the most sensitive components of the land-surface through time. For example, when classifying the occurrence of rill erosion on silt loam soils, the co-occurrence of sparsely vegetated fields (<20% cover) and heavy rain is

a key distinguishing feature (Cerdan et al., 2002), as well as tillage orientation in the downslope direction (Takken et al., 2001). Ultimately, the increased availability spatially distributed data on field parcel structures (Schneider et al., 2023), survey data (Orgiazzi et al., 2018; d'Andrimont et al., 2020), cropping and tillage practices from Earth observation data (Bégué et al., 2018), erosion mitigation measures such as cover crops (Fendrich et al., 2023), and rainfall dynamics (Bezak et al., 2022; Matthews et al., 2022; Kim et al., 2020; Panagos et al., 2023) present opportunities to improve the predictive capabilities of

spatially distributed models. Future spatially-distributed and multitemporal modelling efforts in Europe can be improved in two distinct but interdependent components: 1) validating the temporal dynamics of RUSLE predictions of spatially distributed erosion with plot-scale data to decrease the uncertainty on the on-site predictions, and 2) adding process representation to sediment transport capacity with physical or data-driven models until seasonal bias is minimised (Fig. 10). Both components are arguably key prerequisites to upscaling and necessarily require advancements in the availability of historical instrumental

measurements from small to medium catchments (Matthews et al., 2023b) as well as efforts to compile the available plot data (Maetens et al., 2012).

Despite the outlined challenges, temporally-dynamic model applications give valuable insights beyond the possibilities of temporally-lumped models. These insights can inform continental to global scale model applications on missing driving

processes which justify inclusion. Seasonal vegetation cycles and tillage processes strongly justify treating erosion as a time-explicit process, given that unsustainable levels of soil loss occur during a time-window in which heavy rainstorms coincide with poorly-protected fields (Gabriels et al., 2003). For example, the combination of remotely sensed information on cropping systems with rainfall erosivity time series can identify the on-site erosion risk periods at scale (Matthews et al., 2022; Ayalew et al., 2021; Moller et al., 2017). We further contend that sediment delivery should be considered within a similar temporally

dynamic context, with temporal dynamics potentially diverging from those of on-site erosion. During periods of cropping cycles with low cover, dynamic model parameterisation regimes should consider time-dependent developments in the soil surface condition as well as erosion mitigation measures(Luo et al., 2022; Klik and Rosner, 2020; Azzari et al., 2019; Fendrich et al., 2023; Prasuhn, 2012).

**6 Conclusion**

In this study, we applied standardised catchment-scale WaTEM/SEDEM (W/S) to simulate the multitemporal dynamics of soil erosion and sediment delivery in four catchments in North-West Europe. We show that temporal downscaling needs to be



accompanied by non-linear seasonality in the transport capacity formulation to produce reasonable model performances. The comparison of a temporally-static calibration routine with an optimised multitemporal routine shows that in three out of four catchments, the calibrated transport capacity needs to negatively relate to the predicted RUSLE gross erosion to accurately capture the sediment delivery response. This confrontation of a low-complexity model with measured suspended sediment yield dynamics highlights the need for interdependent progress in: 1) validating multitemporal RUSLE gross erosion predictions, which, in addition to the considered rainfall and crop cover dynamics, may be strongly influenced by antecedent soil conditions, and 2) better representing the time-dependent physical factors which affect the delivery of eroded soil and cause the seasonal coupling and decoupling of gross and net erosion (e.g. tillage state, soil crust development, vegetative barriers, overland flow exfiltration and re-infiltration). Furthermore, temporal downscaling reveals important interlinkages between the temporal scale and the magnitude of catchment spatial fluxes. The implications of a better representation of temporal fluctuations between detachment and transport limited situations impacts the long-term magnitudes soil redistribution, with potential knock-on consequences for secondary application requiring lateral flux estimates (e.g. carbon and nutrient models). In addition to established evidence to show that transport capacity should be structurally independent from RUSLE predictions (i.e. dependent on independent topographic factors), 3 of the 4 modelled catchments show that temporal independence is necessary within the model design. Potentially using the outlined multitemporal optimisation scheme within this study as a proxy to improve the temporal representation of processes, we contend that independent treatment of the spatio-temporal characteristics of erosion risk, runoff, and transport capacity may be necessary for advancements in multitemporal (R)USLE-based soil erosion-sediment delivery models.

## 8 Code availability

The Python code to implement the model input layer pre-processing and multitemporal implementation of WaTEM/SEDEM is made available as a Figshare online repository (WaTEM/SEDEM_dynamic_EU, 2023). Note that all future software and documentation updates will be made through the Github repository (https://github.com/matfran/WaTEM_SEDEM_dynamic_EU.git). The R-factor processing code can be openly accessed online at: https://github.com/cn-ws/rfactor.

## 9 Data availability

All monitored catchment within the EUSEDcollab repository is publicly available via the European Soil Data Centre (ESDAC) via: https://esdac.jrc.ec.europa.eu/content/EUSEDcollab. The relevant example files to run the pre-processing and implementation Python workflows can be additionally accessed via the Figshare repository (WaTEM/SEDEM_dynamic_EU, 2023).



## 15 Author contributions

**FM** Conceptualisation, methodology, software design and implementation, analysis, and all manuscript writing stages. **GV** Conceptualisation, methodology and manuscript review and editing. **PP** Conceptualisation, methodology and manuscript reviewing and editing. **AF** methodology, software design and implementation, and manuscript reviewing and editing.

## 16 Competing interests

The authors declare that there are no competing interests.

## 16 Acknowledgements

The authors acknowledge the financial support provided for this research provided from the Collaborative Doctoral Partnerships (CDP) initiative of the Joint Research Centre (JRC) grant number 35332 and the Fonds Wetenschappelijk
Onderzoek (Research Foundation Flanders -application S003017N). The authors would like to thank Jean-François Ouvry, Jean-Baptiste Richet, Lai Ting Pak and the other contributors to the BRVL and FDTL catchment monitoring campaigns in Normandy, France. For the monitoring efforts of the Kinderveld and Ganspoel catchments, Belgium, the authors extend their gratitude to the efforts of An Steegen, Gerard Govers, and other project contributors. We additionally thank Petra Deproost and Johan Van de Wauw of the Departement Omgeving Flanders, as well as Sacha Gobeyn and Daan Renders from FLUVES
for their contributions to the software developments of WaTEM/SEDEM.

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
