# Peer review of "On the potential of a low-complexity model to decompose the temporal dynamics of soil erosion and sediment delivery"

_EGUsphere, 2023_

## Author Comment (AC1)

**General comments Reviewer #1:**

*"The manuscript investigates the performance of a widely used soil erosion and sediment delivery model (WaTEM/SEDEM) for simulating soil erosion and sediment yield for four catchments in Northwest Europe. The catchment data were taken from a collaborative open-access database and a new open-source model code was implemented in Python. The model, which is usually applied in annual or long-term average annual time steps (as inherited from the RUSLE), is downscaled to a 15-day temporal resolution in order to address the temporal variability of erosion processes in agricultural catchments. Two approaches for model calibration are evaluated: (i) a 'temporally static' one, in which a calibrated parameter set is assumed to be constant during the period of the simulation, and (ii) a 'multitemporal' one, in which two parameters for different transport capacity equations are calibrated on a monthly basis (i.e., one calibrated parameter set for each month of the year per catchment). In both cases, sediment yield data calculated from catchment outlet measurements are used to calibrate and test the model. No temporal and/or spatial split-off testing is employed, and the same data used for calibration are used for evaluating model performance. Both calibration approaches rely on an optimisation function to define a single best-fit parameter set that minimises the differences between measured and modelled outlet sediment yields. The results show an increase in model accuracy with the use of the monthly calibration routine and the authors conclude that this approach improves temporal representation of soil redistribution processes.*

*While I appreciate the general motivation of the manuscript, as well as the use of open-access code and catchment data, I have serious concerns regarding the model evaluation approach employed by the authors. The methods for calibrating and testing the model do not consider the uncertainty in the model or in the input data. Moreover, model evaluation is not performed with independent data (i.e., not used during calibration), which can be misleading. This is particularly problematic for spatially distributed erosion models being calibrated against sediment yield data, as models are able of mimicking outlet sediment flows while misrepresenting internal catchment dynamics. Although spatially explicit soil redistribution rates based on field measurements are available for two of the test catchments, this information was not explicitly incorporated into the model calibration and evaluation processes. Hence, I do not think the modelling methodology is sufficiently sound to evaluate the value of dynamic input data and monthly calibration routines to improve process representation in erosion and sediment delivery models. These issues and several others are discussed in detail in the specific comments below.*

*I also found that the scientific writing is often not precise and that the figures could be largely improved (again, please see the specific comments).*

*Although I see potential in this work, I believe that the changes necessary to address the issues in the manuscript would lead to essentially new results and a new paper, and therefore I cannot recommend this for publication. I do think there is a lot of value in improving the parameterisation of simple erosion and sediment delivery models with dynamic, high-resolution spatiotemporal data – but more calibration (i.e., parameter optimisation) is not the answer, in my opinion."*

**Author general justification of the study approach (all reviews):**

We use this section to respond to general points raised by both Reviewers to address the points in the most concise way possible. We firstly thank the Reviewer for taking the time to examine this manuscript and give both their general and specific reflections on the work. We objectively believe that these comments lead to an iterative improvement of the overall quality of the manuscript. Nevertheless, it is disappointing that the Reviewer comes to the conclusion that the manuscript is unpublishable based on the provided comments. We believe that the majority of the provided general and specific comments are addressable with modifications to the manuscript. We highlight that many of the criticisms made do not concern weaknesses of our work in specific, but issues which are perhaps omnipresent to models of soil erosion and sediment delivery. These include, for example, the challenges in capturing the multitemporal, spatially-distributed properties within catchments using large-scale, open-access datasets with a quasi(-European) coverage, unknown accuracies on spatially-distributed simulations when non-unique solutions are possible, and missing or incomplete (strategic) considerations of uncertainty on the simulated and measured suspended sediment loads in RUSLE-based model applications. We do not contend that systematically improving such issues is not important in future iterations of this work or in other similar efforts, however we do not believe that the specific scientific value of this work is critically undermined by these limitations. We would like to reiterate that the specific goals of the study were to investigate:

*"Can W/S with a temporally static calibration routine decompose the dynamics of sediment delivery from a multi-year aggregation of sediment yield? Secondly, does a multi-temporal calibration routine improve the model performance over a temporally static one? and can the seasonality of posterior output be used to infer the unrepresented processes responsible for the model error. Lastly, the implications of these two temporal approaches were further explored to investigate the interdependencies between space and time on the spatially distributed sediment delivery."*

Which were formulated a specific downscaling exercise of the WaTEM/SEDEM (W/S) model which gives insights into the downscaling opportunities of a semi-empirical model and aids its numerous other applications. With this we aim to identify limitations, such as temporal scaling relationships between gross erosion, transport capacity, and net erosion; and challenge new research directions which can progress this type of modelling at temporally distributed and temporally lumped timescales.

We address the comments raised by Reviewer #1 and #2 firstly by providing a general justification of the approach (since there were questions raised regarding the value of the work), and secondly by addressing the specific comments made. We refer back to this justification throughout the responses:

We acknowledge both Reviewer's opinion that increased input data quality is valuable within these simplistic model frameworks. However, we believe that some of the more fundamental comments arise from a misconception of the approach behind this

manuscript. Firstly, we believe that, within the constraints of practically available data at large scales, improved calibration techniques do have value, including to investigate the potential of multitemporal calibrations to overcome some of the limitations of the low time-dependent process representation in erosion models like WaTEM/SEDEM (W/S). The value comes from (potentially) improved prediction capacity, but also improved understanding of where simplistic models fall short for multitemporal predictions; and, if these have consequences at longer timescales. Given that the W/S model was conceptualised with a long-term annual average time step (~22 year) inherited from the RUSLE, this approach accepts that we are pushing the model to, or perhaps beyond its conceptual limits. Nevertheless, soil erosion and sediment delivery happen at fine temporal scales, therefore the implications of the scaling properties between short and long temporal scales is an important and fundamental research topic (see Ke and Zhang 2024). Secondly, we seek a modelling approach which is practically implementable within a data-to-model workflow, namely, that it can be implemented in numerous European catchments in addition to the four tested ones. This is rare if not absent within equivalent spatially distributed and multitemporal modelling efforts, which typically focus on 1 or 2 catchments (see Figure 11 in the manuscript). This approach also recognises that due to the model sensitivity to input data, the outputs (and potential interpretations) of models such as W/S in multiple catchments will be specific to its inputs (de Vente et al., 2009).

In focussing on data inputs with a European coverage, we also retain a spatial resolution of 25-metres within the model routine, which practically allows the model to make simulations at regional to continental scales (e.g. Borrelli et al., 2017, de Vente et al., 2008). An argument against this could of course be: "why repeatedly implement a model with sub-optimal data inputs if these will lead to a sub-optimal performance?", yet, we believe that there is strong scientific value in progressing modelling efforts which are generalisable based on a standardised schema. Given that models also have use outside of the scientific realm, Schmaltz et al., (2014) gives good motivations for achieving comparable model outputs. This of course in no way depreciates modelling efforts which are specifically adapted to information-rich data inputs, however we strongly emphasise that the interpretations and critiques of this study should consider that we focussed on data inputs with high scalability. Furthermore, the data used for this study is open-access to the research community within the EUSEDcollab database. We would stress that we hold no monopoly over the application of this data in studies applying models with optimal data inputs. Indeed, we believe that such a study would allow a real baselining of the quality of high quality data inputs, and it is one reason for which we make the model code open-access. In fact, these are some of the fundamental motivation factors behind open-access data in that it can aid iterative improvement.

Despite not using catchment-specific data, we believe that with the multitemporal parameterisation of the WaTEM/SEDEM model (event-based EI30 quantifications combined with 15-day soil loss ratio predictions) we are in fact increasing the input data quality over a typical long-term annual average implementation (for example a temporally-static C-factor value based on look-up tables which are often used (Alewell et al., 2019)). Nevertheless, because of the generalised model workflow, we purposefully

do not utilise catchment-specific data, as has been available for multiple years to the research community in the Kinderveld dataset (Van Oost et al., 2005). Using such data would be interesting for a case study addressing the value of high quality data inputs, but in our context we believe it would lead to local specificity which would hinder comparability and the potential of the model to be implemented in other regions. In this regard, we reiterate a key conclusion of Schmaltz et al (2024):

*"The analysis highlighted a clear preference for using national or regional data sets and the use of differing parameterisations, even between model applications using the same model type (USLE). This leads to inconsistent soil erosion assessments, hinders comparison of model outcomes across Europe and potentially enforces inefficient management requirements. This implies that harmonisation may be beneficial in certain cases where a comparison of model predictions is necessary, e.g. large-scale model applications for use in European policy programs such as the CAP."*

We apologise if the aforementioned motivation for our work was not clearly stated in the manuscript and we will make appropriate changes to clearly improve this. We accept from already existing research efforts that sub-optimal data inputs will likely not overcome the challenge of poorly reproduced exact spatial patterns of soil erosion at fine spatial scales. Case studies incorporating improved input data quality (e.g. accurate parcel-specific crop dynamics, tillage practices, cover crops, oriented roughness, and soil surface properties) may present a pathway forward. However, these information components are rare and the authors know of no such methods to reliably quantify them at the large scales. For this reason, we contend that critiques of the manuscript should be made with the consideration that the approach can effectively be automatically implemented in any European catchment with Integrated Administration and Control System (IACS) field parcel data and in which rainfall erosivity can be reliably quantified (in this case, we decided to constrain the hydrometerological inputs to the catchment measurements to reduce error on the hydrological forcings). We believe that this is a large and unique step forward in the practicality of the model implementation in Europe, but we must accept several trade-offs.

The second general point of contention is the evaluation of the temporally-static (routine 1) vs the multitemporal (routine 2) calibration routine. Firstly, we would like to clearly define the logic of using such an approach within the WaTEM/SEDEM model structure. The aim of this exercise is to try to bridge the gap between models which simulate the long-term annual average situation (inherited from the (R)USLE) and those which can simulate the multitemporal dynamics of soil erosion and sediment yield. Routine 1 is meant to be analogous to a standard implementation of W/S, in this respect it can be seen as a temporal downscaling of the European approach of Borelli et al., (2018) on a small catchment sample. We note that redefining the temporal resolution from the long-term annual average (the central limit) to the 15-day variability would represent a major improvement in the current status of a simplistic erosion-transport model available for application at the catchment scale. Accepting that routine 1 performs poorly in most catchments (especially those with considerable sediment delivery in winter), we seek to investigate how a multitemporal ktc within the transport

capacity formulation (routine 2) could account for such deficits. The approach is intended to understand the shortcomings of routine 1, rather than to formulate a model which can predict unknown locations or time periods. Lines 349-352 in the manuscript state:

"We ruled out the possibility of using the model for prediction in unknown locations or time periods given that the calibration time series is not completely distinct from the data used in the validation procedure (Section. 3.3.1). Furthermore, no extra procedures were adopted to prevent overfitting, such as cross-validation or penalisation of the objective function."

With this we intended to make clear that the we are testing if incorporating temporal dynamics into the TC of W/S improves the model performance, but moreover if systematic seasonality arises. Given that we intend to make no spatial or temporal extrapolation of the model, an evaluation procedure based on splitting in space or time to assess the model's predictive capacity is not used. Instead, we chose to use all possible measurement data (motivated also by the Ganspoel and Kinderveld being relatively short) to investigate the model response in the most thorough way possible. In line with the approach, the splines curve describing the monthly ktc_high and ktc_low parameters in the model has 5 parameters which are extracted from the suspended sediment load record alone, thus its complexity is limited. We emphasise that the method in routine 2 does not involve directly finding one calibrated parameter set for each month of the year per catchment. The multitemporal calibration is not directly fitting a monthly ktc value, but a curve describing their shape which prioritises seasonality over random error. This aligns with the low-complexity concept of W/S, considering the existence of 24 15-day intervals in a year for ktc_high alone. In contrast, directly fitting these values would likely result in a nearly-saturated model in routine 2 with a complex temporal profile for the ktc values, which is not really the case. In this way, we accept that calibration is correcting for time-dependent shortcomings (which we seek to interpret to learn about the system itself) and are therefore using the model as an interpretative tool rather than a predictive tool for unknown locations or time periods.

We agree with the further comments raised by Reviewer #1 about whether the ktc optimal seasonality is compensating for error in the RUSLE module or missing processes (please see our specific point-wise comments in response to this). We believe that insights of this kind are constructive in improving modelling efforts of this kind and also for identifying errors in standard (long-term annual average) applications of W/S. However, the idea of a perfect model for the erosion rates is unrealistic, and compensating for missing processes with numerical calibration is a problem that occurs not only in our model but in any attempt to represent the transport of sediments. Within a low-complexity model such as W/S we would argue that such calibration is unavoidable. In order to clarify this matter for the Reviewer and the general reader, we will adapt the manuscript discussion to reflect this. We believe that routines 1 and 2 within a 15-day modelling scheme add valuable insights into the shortcomings of the model beyond what can be achieved when the prediction target is the long-term annual average sediment yield. For this reason, we believe that within the conceptual

framework of W/S and its low process consideration, the multitemporal application represents a step forward which compliments the typical long-term application of the W/S model.

Regarding the comment on the general writing style, we will revise the manuscript and increase the scientific readability accordingly. Please see the specific comments below:

Specific comments

*L40: Increased in comparison to which baseline?*

Author response 1: We thank the Reviewer for this comment. This sentence should refer to a baseline with undisturbed native vegetation cover and geological rates of erosion, as formulated by Montgomery (2007). We will add this accordingly.

*L48-50: Perhaps "began being developed in the 1930s" would be more precise.*

Author response 2: Thank you, we agree. This will be added.

*Please notice that the references for the USLE and the RUSLE are missing in this paragraph. Moreover, the use of the model names and acronyms is not consistent or not defined. For instance, you could consider rephrasing to "[...] the popularity of the USLE and its revised version, the RUSLE (Renard et al., 1997) [..]".*

Author response 3: Thank you, we agree. We will rephrase this section accordingly.

*L59: Can you give an example of these non-linear internal dynamics?*

Author response 4: Thank you for the point. We will add examples which better clarify this sentence. Our intention was to briefly outline why a focus on the long-term annual average poorly represents the nature of the problem of soil erosion. The cited literature by Gonzalez-Hidalgo et al., (2009 & 2012) goes into detail on how soil erosion and sediment delivery is time-compressed into a relatively small number of events, while Kim et al., (2016) highlights that the sediment yield response from individual events is highly controlled by both the external and internal geomorphic variability. Within these individual responses, it can be reasonably assumed that nonlinearity governs the process rate response. Ultimately justifying why the system dynamics require attention. We will resolve the vagueness of the current sentence and better reflect this.

*L65: I understand 'dynamic timescales' as timescales that change. Is this what you are trying to convey? Or finer and/or different timescales? Maybe it would be good to define this somewhere.*

Author response 5: Thanks for the comment. We will better convey this point to state that we mean to timescales which are governed by the process-rate variability. We are focussing on the multitemporal variability rather than predicting a central limit (i.e. the average situation) using W/S, which accordingly inherits its conventional timestep from

the (R)USLE model. The argument is that we miss much management-relevant information on the nature of soil erosion when using temporally lumped models. Coincidently, USLE-based models are the most commonly used for policy-related decisions in Europe (Schmaltz et al., 2024) which acts as an external incentive to include temporal dynamics into such models. We will better adapt this wording to reflect that we are specifically referring to a temporally-lumped (long-term annual average) situation versus one that explicitly considers the temporal variability (e.g. a short period of active erosion).

*L65: Which phenomena?*

Author response 6: Thanks for the question. We will change this to 'the driving and resisting factors governing soil erosion'. We accept that a consideration of the full suite of processes is not given in this study, rather a subset of the factors considered most influential (as determined by the RUSLE). However this sentence is meant to reiterate the point made in the previous response (AR5). We will revise the paragraph accordingly.

*L68: What is meant by soil-erosion dynamics here? Are you talking about processes? Temporal variability?*

Author response 7: We mean the process rate variability at the aggregated-event timestep, as was investigated in the cited literature. We will revise this sentence accordingly.

*L70: Why deterministic?*

Author response 8: A deterministic model is one in which the variables' initial values are combined with a set of equations describing how the variables change to result in model outputs. While this may compete with the acceptance of uncertainty and the existence of stochasticity, we highlight that most modelling frameworks applied in soil erosion and sediment delivery modelling are mechanistic and deterministic approaches. Here we do not intend to question the merits of a deterministic vs stochastic approach to a modelling task of this kind. We also believe that future developments of this work may benefit from the inclusion of stochasticity in the input variables and parameters. Moreover, the advantages, shortcomings and limits of the deterministic (R)USLE model are relatively well understood (Nearing 1998).

*L78-80: I do not see your point here. Models should not be complex because they are tested against outlet sediment yields? Wouldn't it be better to strive for better testing data?*

Author response 9: Thank you for the question. The most frequent procedure for model calibration and validation indeed relies on catchment outlet data. The cited review given by Govers, (2011) outlines the philosophical limitations of using outlet data for optimising complex, spatially-distributed erosion models. The associated discussion around the subject rightfully deserves a lengthy explanation which for the sake of brevity we do not enter into. Nevertheless, yes, increasing the quality of testing data

may be a potential solution for this, such as the inclusion of spatially distributed erosion rates. Yet this data is the most time-consuming and therefore rare. We agree that this would be a valuable goal to strive towards, yet within the scope of this paper we rely on a subsample of data from EUSEDcollab (Matthews et al., 2023), which systematically collects the discharge and sediment yield outlet data. In this study we therefore used the outlet data as the model target, as is typical within the W/S workflow.

*107: Based on the topics and references you covered above; do you believe it is scientifically sound to validate a spatially distributed erosion model based on outlet sediment yields? Perhaps using terms like "tested" or "compared" would attenuate the issue. See Beven and Young (2013) and Oreskes (1998) regarding modelling semantics.*

Author response 10: We will remove the term 'validated' and replace this with 'compared'. The goal (i.e. the predictive target) of W/S is to predict sediment yield, however this requires spatially-distributed input information to describe the critical differences between the erosion susceptibility of land-use types to do so.

*109: This sounds very Bayesian. What is meant by posterior here? Why not just outputs?*

Author response 11: We used posterior to refer  to interpretation of the inputs and outputs after the modelling phase, as was done in Figure 7 of the manuscript. We will remove cases of 'posterior' within the manuscript to remove confusion with a Bayesian approach, and replace this with 'model' outputs.

*L109-110: I think the research question could be more precisely stated, e.g., "how accurately can WS simulate 15-day sediment yields with a temporally static calibration?"*

Author response 12: Thank you. We will amend this.

*L118: Who suggests < 10 $km^2$ for this? For instance, I am looking at Fig. 4 from de Vente and Poesen (2005) and their conceptual model includes gully and bank erosion, as well as floodplain deposition, at smaller scales than 10 $km^2$.*

Author response 13: The threshold of 10 $km^2$ is one given in Fiener et al., (2019) as a threshold beyond which processes in river systems may confound the signal of sediment delivery, however we apologise for not citing this as an example. We will however correct this sentence given that it is not the appropriate threshold to exclude gullying, mass wasting, bank erosion.

*L140-143: I found this exclusion poorly justified. I think one of the main reasons for employing dynamic, continuous-simulation erosion models with finer temporal resolution is precisely to represent extreme, episodic events. This is because a single extreme event can dominate the sediment load signal in small watersheds for several years (Fiener et al., 2019). Hence, the ability of models to simulate such high magnitude low frequency events (even if lumped within a given time period) should be scrutinised.*

*Moreover, based on your rationale for excluding the extreme events, shouldn't the event in March (I can't tell the exact month from Figure 2) 1999 also be removed from the Kinderveld catchment dataset?*

*By the way, I think Figure 2 could use some work. The legend could be placed outside the upper panel, as it is applicable to all panels and the legend symbols can be confused with the actual data points. More importantly, I strongly suggest getting rid of the pie charts (see this blog entry https://www.ataccama.com/blog/why-pie-charts-are-evil).*

Author response 14: This was in fact an error in the writing of the manuscript. In checking over the code it was realised that the March event was not included because there was in fact no precipitation data included in the Kinderveld record for the event due to equipment failure. No rainfall erosivity value was therefore calculated which would trigger a model simulation within the routine. The manuscript of Steegen et al., (2000) mentions an estimated sediment yield but no rainfall records are given for the event to allow a quantification of the EI30. For this reason the event wasn't simulated. We apologise for the confusion and we will amend this.

With regards to the pie charts, to give a quick overview to the reader of which seasons proportionally dominate the total sediment yield we believe that they are fine to give a quick oversight. These are auxiliary to the quantitative plots above in Figure 2 with only 4 categories.

*Figure 1: What are the grey-scaled rasters in the figure? I reckon these are catchment DEMs, though they seem to be missing from the legend.*

Author response 15: These are indeed DEM overlays. We will add a legend to each.

*L164: Why hybrid?*

Author response 16: This was intended to emphasise that the RUSLE model is not purely empirical in its design, since components have physical meaning so it's often termed a grey-box model. We will however remove this to avoid confusion

*L186-187: How exactly is the parcel connectivity implemented? Does this mean a percentage of the sediment is dropped at the field borders? Or it only affects the contributing area/flow accumulation and thus the LS factor?*

Author response 17:  All connectivity-related parameters maintained their standard definition and implementation as within the original WaTEM/SEDEM model code. We will add slightly more explanation of what each parameter does. To find a more in-depth explanation, please see the W/S documentation: https://ees.kuleuven.be/eng/geography/modelling/watemsedem2006/manual_watemsedem_122011.pdf

*L198-199: I haven't looked at the code yet, but this sounds great. Thank you for sharing the model code!*

*L200-210: "Single optimised/calibrated parameter pair" makes me worried… See the Beven (2006) reference you cited above.*

Author response 18: We implemented the model calibration for the temporally static ktc parameter pair according to the typical strategy for W/S. Typically a fixed ratio between the ktc parameter pair would be assigned in a case in which multiple satisfactory responses are simulated (see Alatorre et al., 2010 for a discussion on this). This comes also with the purpose of fixing the spatial patterns of sediment transport deriving from ktc when seeking a global parameter set for multiple catchments. In this case, we did not seek a global calibration strategy so we allowed the ratio of the values to vary and presented their most optimal values (Table 3). We will include this in the text as a consideration and add the relevant literature references.

*Figure 3: Are the RUSLE factors considered parameters or variables?*

Author response 19: We apologise for the terminology issues here. To reduce the ambiguity in the nomenclature used, we decided to adopt the term 'parameter' for the W/S input values that are temporally calibrated in our work, and 'variable' or 'factor' for other input data that are not affected by our calibration procedure (e.g., the (R)USLE). We will modify the manuscript accordingly to clarify for the reader.

*L243-244: According to Van Rompaey et al. (2001), the ktc parameter "can be interpreted as the theoretical upslope distance that is needed to produce enough sediment to reach the transport capacity at the grid cell, assuming a uniform slope and discharge". Based on this definition, how do you interpret these differences in magnitude in the calibrated ktc values for the different TC equations? Moreover, what was the parameter space sampled during the optimisation procedure?*

Author response 20: The interpretation will vary depending on the TC formula being referred to. However, for the generalised transport capacity formulae (TC2 in the MS) which incorporates the topographic slope and drainage area, the definition accordingly changes. Within TC2, Verstraeten et al., (2007) state that ktc:

*"….reflects landscape characteristics that influence sediment transport, such as rainfall intensity, soil erodibility and vegetation, and landscape characteristics that influence runoff generation."*

We note that only TC2 was used for the multitemporal optimisation (routine 2), so this remains the definition for the multitemporal calibration.

Regarding the optimisation procedure, we used the Nelder-Mead method, which is not based on sampling from the parameter space (like simulated annealing or random search would do, for example).

*L251-252: This is a great improvement on the model! Could you add another sentence briefly explaining how this diffusive deposition is simulated?*

Author response 21: The change to the source code was made in the same way as that outlined in Van Loo et al., (2017). As stated, the maximum deposition rate is limited to 5 mm per cell which forces the continual routing of outstanding sediment mass. We will include a sentence to elaborate how this was done and guide the reader to the cited literature.

*L255: I think there is a word missing here.*

Author response 22: Thank you, it should be 'into', we will add it.

*L256: Is the SLR also an independent variable for the dynamic model application?*

Author response 23: The EI30 is typically referred to as the independent variable within the (R)USLE regression model since the other factors scale the response. We recommend visiting the review of Kinnel et al., (2010) to understand this rationale.

*L259: Where is this information given in Table 1?*

Author response 24: The table citation was used to reference back to the catchment reference literature. However, Table. 1 can also include the rain gauge measurement information for each catchment. We will add this.

*L262: Why is EU coverage relevant here? Do you wish to test the model or the model + the EU-available data?*

Author response 25: Please see the general motivation (above). We aim to test the model with EU-scalable data. The approach is based on non-specific data inputs, which allows the model to be run for the different measured catchments in a standard way. We highlight in the manuscript that at this stage the rainfall forcing data needs to be the specific catchment data, which is the only catchment-specific data component, representing a future research task to overcome. Otherwise all data inputs are available across the EU, or are expected to achieve European coverage in the coming years (i.e. the IACS field parcel data). We believe that even with a small sample of catchments (4 in this case), a standard approach is critical to allow quantitative comparisons of the model outputs since W/S will be sensitive to the characteristics of the input data (see de Vente et al., 2009). Secondly, we aim for an approach which can be implemented on more catchments in the future across Europe, which is the motivation behind using a low-complexity and practically implementable model for this task. We will make the motivations for EU data coverage more clear within the MS.

*Table 2: How are the field parcel data incorporated into the model and to the PTEF/Parcel Connectivity parameterisation? How are roads, paths, and field borders represented with a 25 m spatial resolution (assuming the model spatial resolution is being inherited from the DEM)?*

Author response 26: The implementation of these landscape features is made according to the typical W/S method (for the sake of brevity here we refer to the documentation

https://ees.kuleuven.be/eng/geography/modelling/watemsedem2006/manual_watemse
dem_122011.pdf). As in the standard workflow, a field parcel is a unique entity with a
connectivity and PTEF representing each border. Since the model is run at a 25-metre
resolution, these features assume the equivalent resolution on the land-use raster layer
of W/S. For this reason, the model requires a specific calibration for these feature
characteristics which emphasises the need for standardised data inputs (Verstraeten
2006). We will add these points to AR17 when modifying the manuscript.

*Table 2: Is there no information on crop management (crop rotation, tillage type and
orientation, etc) per field parcel in the catchments?*

Author response 27: Please see the general motivation. The data is not consistently
available for all catchments so we do not consider it in the formulation of the model
input variables. We do however use the data for the interpretation of the model results,
in which the optimal ktc values through time are plotted against the surveyed
characteristics of the Kinderveld catchment (see Figure. 8 in the MS).

*L271: Wouldn't temporally dynamic parameters be considered variables?*

Author response 28: Thank you for this. We will consistently change the language to
distinguish between input variables and model parameters throughout.

*L271-275: Where/when do you use the annual (or average annual?) C factor? I imagine that
for the 15-day resolution model you use the SLR as input. This needs to be clarified here.*

Author response 29: We apologise if this caused confusion. The purpose of equations 6
& 7 were to show that the model contains a temporally-decomposed C-factor variable.
That is, 15-day soil loss ratio values in place of a temporally-lumped C-factor value. We
will clearly explain this reasoning, which is intended to show how the temporally
dynamic implementation of WS relates to the long-term annual average model version.
The annual value is not used and we will remove instances of the word C-factor
throughout the manuscript and replace them with the soil loss ratio.

*L280: The SLR is not a decomposed C factor, is it? It is a soil loss ratio, as you explain.*

Author response 30: A decomposing the C-factor (equation 6) results in pairs of 15-day
soil loss ratio and 15-day rainfall erosivity (EI) values. Given that WaTEM/SEDEM is based
on the RUSLE, we frame the formulation as a decomposition of the C-factor. We will
explain in the manuscript more clearly that the SLR is a necessary component for the
formulation of the C-factor. We accept that using the method presented, the estimated
SLR value has some independence from the RUSLE model because while the C-factor
would typically be determined from plot survey information, here we used time series
data from Landsat. We will therefore use the SLR as the correct terminology to describe
the model input variable.

*Equation 7: I am surprised to find out that this relation is crop-management independent –
did I understand this correctly? This would mean that, for instance, the soil loss ratio for a*

*conventionally tilled potato field would be equivalent to mulch-tilled wheat, if both have the same fractional (canopy!) cover, which in your case estimated from again a crop-independent NDVI relationship. Such an approach would introduce a lot of uncertainty to the model parameterisation, which would need to be represented/quantified, particularly during calibration due to equifinality issues – do you agree? For instance, in Germany, the SLR for a soil cover < 10% ranges from 0.08 to 0.94 depending on the crop and tillage type (Schwertmann et al. 1987). What is the land use and what are the typical crop rotations for your catchments?*

Author response 31: We agree that this is an important source of error, which is likewise present in any spatial assignment of a long-term C-factor value which doesn't include parcel-specific management information. Nevertheless, including such information would allow a better estimation of the SLR value through time for each field, but resolving this requires very specific management information which is not available outside of highly extensively surveyed catchments. The Kinderveld and Ganspoel are examples of such a monitored catchment with complete open data, however this is the only catchment as such which the authors know of. The SLR will be sensitive to factors beyond the fractional vegetation cover, such as its structure, the crop row spacing, and as rightly mentioned, the soil management processes.

We use 15-day predictions of vegetation cover based on Landsat data to define the C-factor which is needed to consistently cover the time periods of each of the 4 catchments. We highlight that this is a remote sensing approach to estimate the C-factor, which, as stated in lines 284-287, focus principally on the green vegetation component of the C-factor. The authors know of no equivalent method to predict crop management using Landsat data covering the catchment measurement periods. The field parcel boundary geometries are based on IACS field parcel data from 2018, available in the Eurocrops repository (Schneider et al., 2023). The declared main crop cultivations are therefore more modern than the sediment yield measurements.

We agree that equifinality is an important consideration in any model making spatial predictions using a calibration/target which is not spatially distributed. A large number of factors may contribute to uncertainty on the spatial predictions, and further work which investigates these potential uncertainties in a multitemporal framework would be highly valuable. Nevertheless, within this modelling approach we do not dismiss the issue of unaccounted for management practices. In fact, we aim to interpret these errors at the catchment scale via an analysis of the multitemporal dynamics of the ktc_high factor from the optimal model runs (Section 4.2 of the MS). Without explicitly addressing equifinality in the modelling routine, we therefore did make considerations for the unconsidered management practices relevant for the C-factor. We will however add further discussion on the impact of unconsidered management practices on the accuracy of spatial predictions and the issue of spatial predictions. We will also add a short description of the land use and crop rotation information for each catchment where possible based on the available catchment

*L298: This demonstrates that the NDVI is a good predictor for vegetation cover, which indeed it should be, right? But how does this relation evidence a good correspondence between predicted and observed crop dynamics?*

Author response 32: Indeed there is perhaps little novelty in stating that NDVI predicts vegetation presence through its vitality. However, integrating time series data to describe the seasonally changing SLR values (instead of a C-factor assignment based on a singular temporal acquisition) is a novel application within W/S, which is much better suited to seasonally changing cropping cycles. We would moreover argue that predicting the multitemporal dynamics of vegetation using harmonised Landsat data (especially for historical years in which we have monitored catchment measurements) is a less trivial task, and certainly at most rarely undertaken across soil erosion modelling efforts to the authors knowledge. This is exacerbated given that monitored catchment data is typically historical, and only Landsat-5 data can be used for time periods before April 1999, which for example covers the majority of the Kinderveld and Ganspoel datasets (see Table. 1 of the MS). Sentinel-2 data is much better suited to the task of predicting crop phenology but the data is significantly more modern than the catchment data.

To validate the predicted canopy cover dynamics at the catchment scale, the catchment average SLR value is compared with independent survey data from the Kinderveld to evaluate how well the temporal dynamics of vegetation in the catchment were predicted. These dynamics are dominated by arable cropping cycles, therefore we consider a correction through time at the catchment scale as representing the cropping cycles at an aggregated spatial scale. We do not consider the input data sufficiently accurate to capture the vegetation dynamics for each specific field parcel, however we show that the average crop dynamics are captured using multitemporal Landsat data. We agree that naturally this method could cause spatiotemporal error in highly dynamic field parcels which is hard to compare against a 'true' value. Nevertheless, at the catchment scale we can have more confidence that the average temporal profile of green vegetation is predicted.

*Figure 4: I found this figure very confusing. The solid lines, which are missing from the figure legend, are hard to visualise. The legend inside the upper panel is again confusing.*

Author response 33: Thank you for the suggestions. We will move the legend outside of the panel and include a description of the coloured solid lines in the legend. These are examples of the SLR through time in a sample of 20 individual field parcels from each catchment which we believe are valuable to see the variability in the Landsat derived time series. We will improve the legibility of these profiles.

*L314: Does this mean you performed a temporal split-off test?*

Author response 34: The procedure for defining the calibration target varies between the temporally fixed ktc optimisation (routine 1) and the multitemporal procedure (routine 2). Lines 314 to 321 refer to the evaluation phase which is common between

both routines. Please see the general justification for a full explanation and justification of the calibration procedure.

*L322-332: I had a hard time understanding the calibration procedure. I suggest reformulating so that the methodology is clearly and simply stated for the reader (e.g., what are the parameters actually being calibrated, for which temporal resolution, what data are used to condition the ktc parameters, what kind of split-off testing is employed – or not – and so on).*

Importantly, if I understood correctly, what you call "different connectivity scenarios" are actually part of the parameter optimisation procedure, in which you calibrate the trapping efficiency and parcel connectivity parameters, at least according to the supplementary material (S1.5). This critical information needs to be explicitly stated in the manuscript.

Moreover, I can't say I understand the part about the sediment delivery ratio (SDR) thresholds, which, again, seems like important information that should be clearly explained in the manuscript – not in the supplementary material. In any case, this approach seems to rely on catchment-lumped SDRs calculated from RUSLE-estimated gross erosion rates and measured outlet sediment yields. This seems to assume that the RUSLE-estimated erosion rates are somehow true, so that deviations from expected delivery ratios would be caused by parameterisation errors or the occurrence of other processes than rill and interill erosion (lines 125-135 from the SI). I am not sure I agree, as this assumption apparently neglects (RUSLE) model error, which can be quite large given the discrepancies between the data/purpose the model was developed with/for and the settings where it is being applied. Hence, using this SDR thresholds as part of the conditioning process does not seem prudent to me, at least the way this is currently justified. Maybe I misunderstood something, which in any case is not optimal

*Author response 35: Thank you for the suggestions. Firstly, we will revise both sections 3.3.2 and 3.3.3 to elaborate the explanation of the calibration procedures and make them clearly understandable to the reader.*

Secondly, the intention behind considering different 'connectivity scenarios' was twofold: 1) to not assume that the best connectivity parameters in a dynamic modelling situation would match those default parameters typically applied in a long-term model application, 2) to investigate how different transport capacity formulae perform relative to each other, considering events with different ratios between the predicted gross erosion and measured net erosion (the sediment delivery ratio: SDR). We accept that this however adds considerable complexity and length to the manuscript beyond what can be addressed within the scope of the study, but we note that as with any W/S modelling study that these considerations will always be present. To give a proper explanation and discussion is not feasible for this manuscript. Consequently, we intend to remove the section on calibrating the connectivity parameters within W/S. As rightly pointed out, treating these as fitted parameters adds considerable dimensionality to the model and in our opinion adds confusion to the MS. An arguably strong benefit of W/S

is low parameter dimensionality which we believe is best maintained in his application. Within a typical W/S modelling routine, only the channel positions would typically be defined and the further connectivity parameters would be left as default.

*L334: What about the trapping efficiency and parcel connectivity parameters? And the SDR thresholds? It seems misleading to state that only two parameters are being calibrated.*

Author response 36: Please see the response to the previous comment. We will restrict the calibration routine to only the ktc parameter pair in order to conform to the typical calibration routine of W/S.

*L335-336: Sounds like quite a magical parameter! How does this compare to the definition of the ktc parameter stated above?*

Author response 37: We would say multifaceted rather than magical. The definition (stated above) emphasises the importance of runoff generation, which is not explicitly treated in W/S. A simplistic model such as W/S makes no explicit separation of the multiple factors which influence the efficacy of sediment transport through time. Therefore, the need to explicitly state that multiple confounding factors will influence ktc in different ways through time is an inevitable consequence of not explicitly treating physical processes within the model. We believe this is reasonable, and an in-depth discussion to accompany this rational was given by Prosser and Rustromji (2000).

*L336-339: Great to have an interpretation of the results, but do you think this is enough to open the calibration black box? You risk affirming the consequent without additional independent and spatial data to support your interpretation.*

Author response 38: We believe that value is certainly added beyond the typical temporally-lumped calibration of W/S. We reiterate that the W/S model was not chosen due to its conceptual suitability to the task of modelling soil erosion and sediment delivery at fine temporal scales. Rather, given that the model is the most widely utilised model for catchment-scale, spatially-distributed modelling efforts (Borrelli et al., 2021), there is considerable value in testing the model at various temporal scales. Certainly, identifying systematic seasonality in the calibrated ktc parameter gives valuable scientific insights which complement the most common temporally lumped model application at a long-term annual average timestep. Without disaggregating the individual effects of the contributing factors to the ktc (which would likely require high spatially-distributed detail), we can instead add parameters which emphasise the temporal component of ktc and therefore make inferences about dominant processes. Therefore, we believe that this kind of calibration is beneficial, and we make an effort to confront the interpretations with the observations made in the modelled catchments to give an explanation of the model outputs. We would therefore call it illuminating the black box, which is an approach compatible with such a semi-empirical model.

*L340-350: So, the same data are used for forcing and testing the model? This hardly seems justifiable, considering the temporal and spatial data available for your catchments. These*

*data would allow for different types of split-off tests and for an evaluation of the transferability of the calibration procedure.*

Perhaps more importantly, why did you not account for the equifinality issue during calibration? Even with a small number of parameters being calibrated (which I am not sure is the case here), there are several model realisations able to mimic the outlet sediment data if we consider the degrees of freedom afforded to spatially distributed models, and the errors in the input and forcing data – particularly with a monthly calibration. All of these well-known issues, as well as methods for addressing them, are described in some of the references you cite in the manuscript.

Moreover, why didn't you use the spatially distributed erosion data from the Ganspoel and Kinderveld catchments to calibrate and/or test the model? The data were specifically collected for this purpose, as stated in the title of Van Oost et al. (2005).

Author response 39: Please see the general reasoning above which justifies the model approach. We believe that this comment relates back to the two main points addressed in the general justification of the model approach. Namely: 1) the justification for not using all catchment-specific data within the workflow in order to create a model workflow which is generalisable to other catchments, and 2) the justification for not performing a spatial or temporal split of the suspended sediment load data given that no attempt was made to extrapolate the model. The latter of course accepts that we are not seeking an out-of-the-box solution to modelling the 15-day sediment load using W/S, rather we seek to understand if there is systematic seasonality within the optimal ktc profile. Lines 351-352, describing the multitemporal optimisation routine of the ktc profile, state:

*"Furthermore, no extra procedures were adopted to prevent overfitting, such as cross-validation or penalisation of the objective function."*

The intention of this statement was to make clear that the method was not formulated for extrapolation, so therefore was not used as such. as stated in lines 348-351, the objective function in the case is the root mean squared error on the mean monthly average sediment yield, therefore a statistical reduction of the full suspended sediment load time series. By using a statistical reduction of all data we intended to have more certainty on the average seasonality required in the ktc parameters based on all available observations, with the purpose of interpreting deficits within the W/S model rather than making predictions in unknown locations. The model evaluation was then undertaken on all individual 15-day suspended sediment loads. Firstly, without penalising the shape of the objective function, the curve determining the monthly ktc value can fit relatively freely to minimise the objective function. Secondly, the absence of a cross validation (i.e. the superimposition of the splines parameters describing the ktc curve from one catchment to another) means that the multitemporal ktc profiles are catchment-specific. Therefore they are specific rather than global parameters, which become useful for interpretation rather than extrapolation. We will modify the methods section to clearly explain this and avoid confusion for the reader. However, no effort is

made in the manuscript to say that the model result is satisfactory for extrapolation. Indeed, far more attention was given in the manuscript to the interpretation of the multitemporal parameters rather than the model results.

With regards to the equifinality, we acknowledge that this issue is inherent within any modelling effort employing a spatially distributed model calibrated on the catchment outlet. The implementation of the W/S model within an optimisation software (scipy.optimise) was to the authors knowledge the first open source implementation of an efficient parameter estimation procedure, such as that used in Bezak et al., (2015). Moreover, we show the applicability of the method to minimise a function describing the multitemporal profile of calibrated parameters. In this case, we limit the multitemporal calibration to only the ktc_high and ktc_low parameters. Nevertheless, the necessary model iterations to minimise the objective function were typically over 100 runs of the model at a 15-day resolution, taking between 5 hours and a day depending on the catchment size on a 11th Gen Intel Core i7-11800H processor at 2.30GHz using 32 GB of RAM. A rough calculation, assuming 100 iterations to minimise the objective function, means between 1800 and 5200 runs of the W/S model per catchment. Repeating this to allow variation in other parameters becomes rather computationally costly depending on the method and parameter space sampled. Such an exercise would without doubt be interesting, however it would mean a significant upscaling of the computation. We will modify the manuscript discussion (section 5.2.2) to emphasise the remaining uncertainty due to the issue of equifinality in spatially distributed models.

The spatially distributed data are not explicitly considered because sub-optimal data are used for the parameterisation scheme with justification. Given the non-specific modelling workflow, we purposefully do not use the spatially distributed information to formulate the model input variables. Figure. (8) shows an example of the catchment survey information used in a spatially lumped way to interpret the temporal dynamics of the ktc parameter. A direct pixel-to-pixel statistical analysis was not considered appropriate, due to sub-optimal input information, which includes for example a 25 m DEM which does not resolve complex topographic detail. We will however add a further analysis for the Kinderveld catchment which evaluates the spatial accuracy of the erosion predictions. Accepting that a pixel to pixel analysis would not be appropriate, we intend to perform this with an appropriate level of spatial aggregation (e.g. Peeters et al., 2006).

*L352: Why does avoiding a cross-validation prevents over fitting?*

Author response 40: Please see Author response 39 for the details of the splines fitting procedure. We in fact state the opposite, that not undertaking a cross validation leaves the multitemporal ktc profile susceptible to overfitting, and therefore less suitable for extrapolation. We apologise for confusion in the working and we will modify section 3.3.3 accordingly so that this is clearly explained.

*L362-365: If I understood this correctly, the measured sediment loads do not correlate with the simulated erosion rates from the RUSLE for most (3) of the catchments. Hence, it seems like (i) there are other processes not simulated by the model that are affecting the sediment yield or (ii) the model is not fit for purpose. I imagine now that if you include a monthly calibration your results will improve, specially since there was no split-off testing or uncertainty estimation. Do you reckon this means the model improved as a representation of the system or it simply improved its capacity to mimic the forcing data?*

Author response 41: Indeed, for 3 of the catchments, the predicted gross erosion poorly correlates with the catchment net erosion. Lines 520 to 540 discuss this, and we believe that such a result gives interesting insights into where errors can occur when using RUSLE-based predictions to predict the net sediment yield. Our goal in this results section was however not to discuss the reasons for such a result, rather to show that at the 15-day timestep the RUSLE gross erosion poorly correlates with the net sediment yield.

*Table 3: The SDR information could be better explained, I am not sure what SDR (max) means. I would also like to see all calibrated parameters here, not just this lumped "connectivity index" (which sounds too much with other indices used in connectivity research e.g. Borselli et al., 2008).*

Author response 42: As noted above, we intend to remove the study component which considers the further connectivity related parameters within the calibration scheme. Please read Author response 35 for a complete reasoning.

*I am again surprised by the variability in orders of magnitude of the values for the calibrated ktc parameters. Seems like the parameter has been stripped of its original physical meaning and became an adjustment factor to fit the forcing data.*

*What was the parameter space you sampled? Apparently, you gave the model a lot of room to fit the forcing data.*

Author response 43: Please see the previous Author response (39). We did not implement limits in the parameter space, which gives the ktc profile flexibility to find the most optimal possible profile to fit to the suspended sediment load record. Here we are essentially asking: 'what is the optimal multitemporal ktc profile within the W/S model to allow the model to replicate the sediment delivery dynamics?'.

We note that the physical meaning of ktc (author response 20) is not violated since there can be numerous reasons since it is influenced by the confounding of numerous factors which operate at a spectrum of spatial scales.

*L380-385: Yes, as expected we see a "boost in model performance", as you call it. This would be great if it had been achieved by improving the dynamic model parameterisation with measured data. What we see here seems to be the result of an increase in the freedom the model has been afforded to fit the sediment yield data. I assume that if you do a weekly or daily calibration the results would be even more accurate – despite the fact that USLE*

*predictions are known to deteriorate at finer timescales (Risse et al., 1993). Hence, the erosion predictions get worse; but the sediment yields are more accurately simulated – isn't this in principle contradictory for small catchments with a predominance of rill and interill erosion and negligible channel processes (i.e., your reasons/assumptions for choosing the test catchments)? That is, if these assumptions are true, shouldn't the sediment yield be largely explained by the hillslope erosion rates, particularly for 15-day timesteps? In Table 2 we see that the catchments in which predicted erosion rates do not correlate with the measured sediment yields now display the highest NSE values. What does this mean?*

I really think it is a great idea to improve the ktc parameterisation in W/S to account for temporal variability in roughness, vegetation cover, etc. But more calibration without uncertainty estimation and using the same (outlet) data for forcing and testing the model is not the answer, in my opinion.

Author response 44: Thank you for the raised points. We firstly refer back to the general justification in which we did not use specific catchment data as model data inputs. This ruled out the possibility to pursue the approach that the Reviewer considers as a more idealistic approach to improve the representation of ktc. Nevertheless, we believe that given that W/S has a low-complexity, even with more accurate data inputs the issue of confounding factors influencing the interpretation of ktc would remain an issue. For this reason, we focussed on an approach which could use all available data and expose systematic seasonal errors. Important to emphasise is that we seek to understand where the model goes wrong in a temporal-downscaling exercise to give insights into how long-term applications may accumulate error. Given that any long-term quantity of sediment delivery is an accumulation of numerous events of unequal contributions.

Any utilisation of the RUSLE model fixes the prediction target to that which would be expected at the scale of the unit plot. At the scale of a small catchment, it is feasible that the ktc dynamics could significantly differ from the predicted for reasons of: 1) unaccounted for processes impeding or promoting sediment delivery, or 2) error in the RUSLE predictions at the 15-day temporal scale. The employed method is novel because it emphasises seasonality in the ktc parameter rather than random error. Optimising the model based on daily or weekly ktc values would therefore risk having the opposite effect. Secondly, the ktc multitemporal profile is described by 5 parameters and therefore in a hypothetical situation in which specific daily or weekly ktc values were input into the model, these would still be determined by a splines curve with a complexity limited to 5 parameters. We instead believe that exposed seasonality within the ktc value, or transport capacity, is in fact interesting in the context in which it is applied.

We accept that RUSLE is typically used as an out-of-the-box model due to its design, which, without explicit spatial validation of its target variable (the plot-scale sediment yield) at the appropriate spatial (i.e. the unit plot in the landscape) and temporal (in this case at the 15-day interval) scale can induce unknown errors into the gross erosion predictions. Unfortunately, data to perform such a like-for-like validation task is rare, and not present for the modelled catchments. This, consequently means that

interpretation of the ktc multitemporal cannot be seen independently for the RUSLE parameterisation regime and its performance at fine temporal scales. Lines 520-540 within the discussion intend to shed light on this, which we believe adds value into where long-term applications of W/S may accumulate error.

Regarding the Nash-Sutcliffe efficiency (NSE), that is an interesting question. We did not strongly interpret the differences in the NSE between catchments due to the sensitivity of the NSE to the catchment sediment load dynamics (Krause et al., 2005).

*L398: I would not say there is evidence that this calibration corrects any errors – it might simply compensate one error with another (e.g., Pontes et al., 2021).*

Author response 45: Thank you for the point. Indeed we did include this in the discussion as a key take-away message on lines 466 and 467 and in more depth between 528 and 540 as we believe that this is an interesting interpretation which can be made. We will revise this sentence accordingly, and likewise provide an extended discussion on why the interpretation of ktc should be made as a potential error compensator; particularly in the predicted gross erosion.

*L398-410: One could also say that during summer there is an overprediction of gross erosion rates which is compensated by calibrating the ktc parameter with very low values that increase hillslope deposition. How do the estimated deposition rates and patterns compare with the measured data in Van Oost et al. (2005)?*

Author response 46: We believe that this is indeed possible and we raised this as a point of discussion in section (5.1). Figure (10) shows that this is likewise the case, being that the implemented RUSLE model systematically overpredicts the summer gross erosion or doesn't account for the strong influence of soil crusting in the K-factor during the winter. We will increase the discussion on the possibility of the ktc to compensate for error within the predicted RUSLE model, which we believe reveals interesting seasonality. As mentioned in AR 39, we intend to perform a spatial analysis of the simulated erosion and deposition rates in an appropriate way and add these to the manuscript.

*L415-420: Aren't these additional signs of the model compensating for under- and overprediction of erosion by means of calibration?*

Author response 47: We agree. Indeed, this analysis was intended to show how an analysis of the calibrated multitemporal ktc value alongside the input factors can be useful to infer model error. In this case, this was made without catchment specific model inputs, apart from rainfall data. As mentioned in the previous AR, we intend to formulate a more in-depth discussion on seasonal error compensation and what it could reveal about the RUSLE model.

*L438-440: What does moderate capacity mean? Could you give us numbers please? Moreover, what are we supposed to look at in Figure 9c and d? What is modelled and what is measured there? I found Figure 9 to be little informative and hard to interpret.*

*Importantly, how do the modelled soil redistribution rates compare with the measured redistribution data in Van Oost et al. (2005)? What about the Ganspoel catchment?*

Author response 48: We did not perform a quantitative assessment of the pixel-scale classification accuracy of areas subject to erosion and deposition. This was not performed for two primary reasons: 1) The specific catchment data was not used to parametrise the model, and factors such as tillage orientation were not accounted for within the modelling routine, although they are known to be highly important in dictating the spatial patterns of erosion (Takken et al., 2001), and 2) W/S was run at a resolution of 25 m, which is the equivalent spatial resolution of a regional or continental scale application (e.g. Borrelli at el., 2017). This retains the advantage of the practical implementation of W/S at large spatial scales, but we didn't anticipate the model to resolve the fine scale properties of erosion and deposition which a visual survey can describe. We did however visualise this data for the reader to see how the modelled erosion and deposition compare with the surveyed areas showing erosion and deposition in the Kinderveld catchment (Figures 9c and d). "Moderate" means in a qualitative sense that most surveyed depositional areas (> 0.5 mm deposition) are modelled as such, and likewise areas simulated as having appreciable erosion (> 1 mm erosion) contained surveyed signs of erosion. We will better describe the aforementioned points in the manuscript and also justify why the pixel-scale comparison was kept as a visual comparison.

Figures 9c and 9d show a comparison of the modelled and surveyed erosion and deposition, aggregated over the entire measurement period of the Kinderveld catchment. We apologise if this was not completely clear. These figures compare pixels with simulated erosion (>= 1 mm) and deposition (> 0.5 mm) to pixels with visually observed erosion and deposition in the Kinderveld. We will extend the description of this within a new section of the methods explaining the spatially distributed analysis of the erosion and deposition.

In the revised manuscript we will include a specific method and results section focussing specifically on the spatial estimates of erosion and deposition. We included the Kinderveld as one example of the spatial patterns (Figure 9c and (d). However, given that no quantitative analysis was made, we did not include the Ganspoel data. We will likewise include the spatial surveys of erosion and deposition data for the Ganspoel in the revised version. We will also divide figure 9 into different figures which focus on only the model simulations (9a and 9b), those which compare simulations with the Kinderveld survey data (9a and 9b) and also separate the large figure panel (9e) which compares the spatial magnitudes of erosion and deposition for the temporally static ktc calibration, with those from the multitemporal ktc calibration. As reasoned in this response, quantitative analysis will be limited to spatial aggregations in line with Peeters et al., (2006).

*L465: But the BRVL catchment showed no correlation between SSL and gross erosion predictions (r = 0.04) and, at the same time, the highest NSE values following the monthly calibration... Did I understand something wrong here?*

Author response 49: This section refers to the temporally-static calibration of W/S only. This point was used to emphasise that all catchments with a non-significant correlation between the predicted gross erosion and the suspended sediment load performed poorly (NSE < 0) when W/S was applied with a temporally static ktc value.

The results were as so for the correlation between the RUSLE gross erosion and the in-stream suspended sediment load:

Ganspoel (r = 0.64, p < 0.05) > Kinderveld, (r = 0.23, p > 0.05) > FDTL (r = 0.18, p > 0.05) > BRVL (r = 0.04, p > 0.05)

And the the 15-day

Ganspoel (NSE = 0.28) > BRVL (NSE = - 0.03) > FDTL (NSE = - 0.04) > Kinderveld (NSE = - 0.09)

*Please note, there was a mistake in the ranking order within the manuscript. We will correct this in the revision.

We apologise if this was not completely clear. We will make it clearly in the revised manuscript that this statement refers to the W/S model run in a conventional way.

*L467: It can compensate the error; but does it improve the model's representation of the system? If we would only care about accurately simulating the sediment yield, why would we need a spatially distributed model anyhow?*

Author response 50: We intend to investigate in the most appropriate way how the temporally dynamic model implementation improves on the temporally static version, beyond better replicating the output sediment load. Nevertheless, in the context of interpreting the opportunities and limitations of the W/S model, we believe that there are at least interesting insights into where the long-term annual average version of W/S may misrepresent the system dynamics. We reveal systematic seasonality in the ktc parameter, or transport capacity, of which a significant proportion likely belongs to variance in efficacy of sediment transport throughout the year.

Moreover, spatially distributed models have numerous motivations, which can include the ambition to more closely represent the natural system, to not generalise the spatial patterns of erosion which are spatially concentrated by nature, and to seek to obtain the right result for the right reason. Spatially distributed input data remain crucial to properly describe the sediment yield in a catchment, and moreover the areas contributing most to the sediment delivery. An appropriate description of the spatially-distributed land elements with high erosion susceptibility has been shown to be important in determining both short-term and long-term erosion rates (Peeters et al., 2006). We therefore contend that the methodology is appropriate.

L476-477: Again, these are not measured gross erosion rates.

Author response 51: We will make adjustments to make it clear that in our case the gross erosion is indeed predicted. As it often is due to the difficulties in measuring spatially widespread and diffusive processes across a catchment.

*L481: "Winter runoff events dominate the runoff and SSL budget in the BRVL and FDTL catchments (Grangeon et al., 2022), representing cases in which the seasonal dynamics of predicted gross erosion and measured SSL were inverted."*

*Then doesn't this potentially indicate that (i) the model is not fit-for-purpose for simulating these catchments in this temporal resolution and that (ii) you are compensating the spurious temporal simulations of internal soil redistribution by means of calibration of the sediment yield?*

Author response 52: With regards to point (i), without seeming evasive we would contend that the idea of a fit-for-purpose model depends on the purpose of the modelling effort. The intention of this exercise was not to directly produce the best model fit for extrapolation in space or time. Rather to investigate how W/S, a low complexity model with a practical implementation routine, can downscale to capture the system dynamics at a finer temporal resolution. Without doubt, the W/S will be expected to be less fit for modelling at a 15-day temporal scale compared to process-based models such as WEPP, LISEM or other models resolving physical processes with a significantly higher level of detail. However, the investigation of how a model such as W/S can transverse temporal scales is of high interest. Particularly, since at the long-term annual average temporal scale, the ktc parameter is considered to linearly couple with the gross erosion, i.e. all eroded sediment has an equal transport efficiency. Here, we give useful insights to show that this is not the case, which may have compounding consequences for W/S and similar models applying the widely used transport capacity equation ($TC2 = ktc\,A^{1.4}ST^{1.4}$).

The secondary point (ii), we accept that systematic error compensation on the predicted gross erosion can be a component of the seasonality within the ktc multitemporal profile. Indeed, the design of W/S makes it very difficult to disentangle changes in the efficacy of transport capacity (strongly affected by runoff generation) from plot scale gross erosion predictions (also strongly affected by runoff generation). As previously mentioned, we intend to further discuss these error compensation effects as a component of the ktc multitemporal profile as we believe they give interesting insights into the model.

*L491: I agree, but here you have access to the great spatially distributed erosion data from the Kinderveld and Ganspoel catchments. Why haven't you used them?*

This whole discussion made me think about how the concepts of gross erosion and sediment delivery ratio are somewhat inadequate and how perhaps we would be better off thinking in terms of travel distances (Parsons et al., 2004, 2009).

Author response 53: We intend to apply the Kinderveld and Ganspoel data in the most appropriate way which matches the characteristics of this modelling exercise. The consideration of erosion in terms of travel distances or the conventional gross and net erosion is an interesting point, but one which requires discussion beyond the outlined scopes of this study.

*L545-546: Where is this provided in Table 1? Uncertainty estimation was indeed missing here.*

Author response 54: We provide a list of all literature references attached to each dataset. We will make clear that this table citation is directed to the available literature on each catchment.

*L551: I would argue that in the monthly calibration (which seems to be the precise term here – not 'multitemporal calibration') you define an optimisation routine to mimic the 15-day sediment load data, but you haven't provided evidence this is achieved for the right reasons. In fact, the calibration might be compensating for errors in the model and the model parameterisation (see comments above). Hence, I do not think it is sound to state that this calibration serves as a "proxy for missing parameter information or process components".*

Author response 55: We disagree, given that monthly calibration routine may infer that the calibration process was conceived to directly find 12 (or more) monthly ktc parameters when this was not the case. The calibration routine finds a multitemporal splines curve from which the monthly ktc values are determined for the model. As we previously mentioned, we believe it is important to communicate that the model is not saturated in a sense that we directly found the kc value which fits best per month.

Regarding the second point about the ktc curve serving as a proxy for missing seasonality. Indeed, we do not refute the proposition that a multitemporal ktc can compensate for errors in the multitemporal RUSLE predictions, and we provided a discussion on this (Lines 520-540). In the following sentence we elaborate on this, stating that:

"Hypothetically, the multitemporal optimisation routine would derive a flat ktc curve, equivalent to the temporally static calibration routine, in a situation in which all variability in the response variable (15-day sediment load) were accurately described by the model."

Which would mean that no systematic seasonal bias is present within the predicted sediment delivery. We intend to elaborate this and make it more clear that we are referring to seasonality, which is what we intended to expose in a multitemporal calibration routine.

*L566: I don't really get how you are using the term "deterministic" throughout the text. What is deterministic model performance?*

Author response 56: Please see AR 8 in which we explain this.

*L572: Can you give the reader a brief explanation of this matter here and refer to Steegen and Govers (2001) for details?*

Author response 57: Yes. We will discuss in more detail the potential reasons in which connectivity may explain the better correlation between the predicted gross erosion using the RUSLE and the suspended sediment load in the Ganspoel catchment compared to the Kinderveld catchment.

*L587-588: "Nevertheless, non-linear temporal differences in ktc back-propagate over the landscape to change the magnitudes of erosion and deposition (Fig. 9)." I am not sure I get this. Can you be more direct? To be honest, I did not get were you wanted to go with this paragraph.*

Author response 58: Our apologies for not making this clear. Indeed we will specify more clearly what we mean in the manuscript. Briefly, what we mean is that within the W/S model structure, the most common shape of the multitemporal curve (i.e. in 3 out of 4 catchments) strongly reduced the transport capacity during the summer, in which the predicted gross erosion is highest. This prompts a significantly higher number of cells with transport limitation and therefore reduces the magnitudes of sediment which are displaced across the landscape. Ultimately, given that ktc in W/S is a calibrated parameter assigned across the landscape based on the output sediment yield, we show that considering temporal dynamics results in different magnitudes of sediment redistribution.

The goal of this paragraph is to confront this first 15-day application of W/S with the typical long-term average application of the model. We suggest that the temporal scaling properties of the ktc parameter appear to be non-linear rather than linear, which must have an impact at aggregated timescales. From our analysis of the spatial magnitudes of the sediment redistribution in Figure (9), this results in a decrease in the magnitudes of sediment redistribution. We will revise this paragraph to more clearly emphasise this point, which we believe is a rather interesting interpretation which comes from implementing W/S at the 15-day timescale.

*L610-612: Same here: "The spatial characteristics of soil erosion represent the source of the cascading environmental impacts, arguably making them the key prediction target (Vigiak et al., 2006; Jetten et al., 2003; Merritt et al., 2003). While error on the spatial patterns of soil erosion and sediment transport can confound within the spatially lumped sediment yield, the spatial patterns can remain poor (Jetten et al., 2003)".*

Author response 59: Here we aim to emphasise that like in most spatially distributed models, there is a susceptibility for models to capture the sediment load magnitudes at various timescales while misrepresenting the spatial patterns. We believe there is strong potential for better data inputs to overcome this. Nevertheless, we agree that since we gave limited interpretation of the spatial patterns of erosion and deposition in the manuscript, this discussion can seem isolated. As stated, we will add results on the

spatial performance of the model in the Kinderveld and Ganspoel catchments, and link this discussion with the results of this manuscript more clearly.

*L627: Do you mean at the erosion-plot scale? Moreover, how does model testing reduce the uncertainty in the predictions? Does quantifying model error reduce model uncertainty?*

*In general, I had a hard time understanding where you wanted to go with section 5.3, which seems somewhat speculative and decoupled from your actual results.*

Author response 60: Yes, firstly we mean the plot-scale erosion covering a small but spatially-distributed area. Practically speaking, this means collecting plot measurement data across Europe and attempting to build new models capable of simulating the multitemporal dynamics of gross erosion. As previously mentioned, within the model structure, separating error in the transport capacity from error in gross erosion predictions is difficult; which we believe is a challenging but interesting aspect of a multitemporal calibration routine.

Furthermore, we would argue that model testing is a key part of identifying and treating uncertainty when performing uncertainty quantifications. For example, we accept the possibility for considerable uncertainty in W/S due to the low process representation. However, the results of this study indicate significant temporal variability in the K-factor variable, as well as other soil hydrological properties which hinder and encourage runoff. We know of no W/S study which has considered the impact of multitemporal variation in the input variables, or the potential impacts that this may have on the representativity of the long-term model simulations.

With regards to the general idea behind section 5.3, we aim to highlight the opportunities which come from modelling routines which are standardised and implementable in multiple catchments. We aim to discuss the novelty of: 1) standardised model implementations in small catchments with measured data, 2) the novelty of multitemporal calibrations in low-complexity models, and 3) the need for better spatially-distributed data inputs for scalable modelling efforts such as this one. In fact, Figure (11) shows that spatially distributed modelling efforts which operate at a sub-annual time step remain relatively rare within the research field. Singular catchment models, which focus on the optimal data inputs for their use case are much more common, but issues with inter-comparing the model outputs of such exercises will remain a fundamental hindrance (Borrelli et al., 2020). We accept that more development is necessary to be able to implement the multitemporal calibration of the ktc parameters in unmeasured time periods or catchments, particularly due to the need for future independent validation. Nevertheless, we believe that this approach to better represent the temporal dynamics of sediment delivery has value due to its practicality and the potential to perform catchment intercomparisons.

We accept however that section 5.3 can appear separated from the actual results. In the manuscript revision we intend to include figure (11) in the introduction where we justify

the modelling approach employed here. The further discussion points will be modified to better link with the results produced in the manuscript.

*L646-647: What is non-linear seasonality? In any case, wouldn't it be more accurate to state that "reasonable model performance" (what is reasonable anyway?) was only achieved after a monthly calibration of the ktc parameters? And that the best-fit calibrated parameter set was not tested against independent data (i.e. not used during calibration)?*

Author response 61: Thank you for the comment. We will revise this to words of the effect:

"We show that temporal downscaling needs to be accompanied by considerable seasonality in the transport capacity formulation to reproduce the 15-day dynamics of sediment delivery."

We will further emphasise in the conclusion that no validation on independent data was made, and rather we used all available data to interpret the multitemporal dynamics of the ktc in the four catchments. Among the other mentioned avenues in the conclusion to potentially improve the model for the task of 15-day simulations, we will further emphasise that future work should aim to perform a validation of this exercise using a split-sample procedure.

*L661-664: I strongly disagree that the monthly parameter optimisation procedure suggested here improves temporal process representation, due to all the above-mentioned reasons.*

Author response 62: We believe that this is a misinterpretation of this sentence. We are simply stating that within the current W/S model, significant processes operate at the intra-annual timescale which are not captured by the model. A multitemporal calibration will compensate for these through the ktc factor, but we do not content that we increasing the number of processes. We state therefore that the multitemporal calibration can be used as a proxy for further model improvements, or potentially using a brute force method to improve the model's ability to replicate the mutlitemporal dynamics of the sediment load. Our opinion is that improving such process representation requires a direct consideration of the processes, something which is currently not attempted in this modelling effort. We further point towards the importance of changes in soil surface hydrological properties which we believe to be useful information for improving low-complexity models in loess catchments.

References

Beven, K. J.: A manifesto for the equifinality thesis, J. Hydrol., 320(1–2), 18–36, doi:10.1016/j.jhydrol.2005.07.007, 2006.

Beven, K. J. and Young, P.: A guide to good practice in modeling semantics for authors and referees, Water Resour. Res., 49(8), 5092–5098, doi:10.1002/wrcr.20393, 2013.

Borselli, L., Cassi, P. and Torri, D.: Prolegomena to sediment and flow connectivity in the landscape: A GIS and field numerical assessment, Catena, 75(3), 268–277, doi:10.1016/j.catena.2008.07.006, 2008.

Fiener, P., Wilken, F. and Auerswald, K.: Filling the gap between plot and landscape scale - Eight years of soil erosion monitoring in 14 adjacent watersheds under soil conservation at Scheyern, Southern Germany, Adv. Geosci., 48, 31–48, doi:10.5194/adgeo-48-31-2019, 2019.

Van Oost, K., Govers, G., Cerdan, O., Thauré, D., Van Rompaey, a., Steegen, a., Nachtergaele, J., Takken, I. and Poesen, J.: Spatially distributed data for erosion model calibration and validation: The Ganspoel and Kinderveld datasets, Catena, 61(2–3), 105–121, doi:10.1016/j.catena.2005.03.001, 2005.

Oreskes, N.: Evaluation (not validation) of quantitative models, Environ. Health Perspect., 106(6), 1453–1460, doi:10.1289/ehp.98106s61453, 1998.

Parsons, A. J., Wainwright, J., Powell, D. M., Kaduk, J. and Brazier, R. E.: A conceptual model for determining soil erosion by water, Earth Surf. Process. Landforms, 29(10), 1293–1302, doi:10.1002/esp.1096, 2004.

Parsons, A. J., Wainwright, J., Brazier, R. E. and Powell, D. M.: Is sediment delivery a fallacy?, Earth Surf. Process. Landforms, 34, 155–161, doi:10.1002/esp, 2009.

Pontes, L. M., Batista, P. V. G., Silva, B. P. C., Viola, M. R., da Rocha, H. R. and Silva, M. L. N.: Assessing sediment yield and streamflow with swat model in a small sub-basin of the cantareira system, Rev. Bras. Cienc. do Solo, 45, doi:10.36783/18069657rbcs20200140, 2021.

Risse, L. M., Nearing, M. a., Laflen, J. M. and Nicks, a. D.: Error Assessment in the Universal Soil Loss Equation, Soil Sci. Soc. Am. J., 57(1987), 825, doi:10.2136/sssaj1993.03615995005700030032x, 1993.

Van Rompaey, A. J. J., Verstraeten, G., Van Oost, K., Govers, G. and Poesen, J.: Modelling mean annual sediment yield using a distributed approach, Earth Surf. Process. Landforms, 26(11), 1221–1236, doi:10.1002/esp.275, 2001.

**General comments reviewer #2:**

*This paper proposed applying the WaTEM/SEDEM model to four well-documented catchments and improving its temporal resolution to a 15-days time step to better represent seasonality effects on modelled sediment fluxes. This improvement is made in response to the perceived lack of temporally explicit modelling approaches in soil erosion modelling. To this end, a temporally varying transport capacity is developed and included using a two-step modelling approach: i) the transport capacity is fixed over the simulation period and is then ii) calibrated on a monthly basis. One of the main result is that using a constant transport capacity parameter throughout the hydrological year yield unsatisfactory results, while the inclusion of two time-varying transport capacity parameters significantly improved model performance.*

*The study relies on the use of open access data, used for model calibration and evaluation, to improve an open-access model. Indeed, one of the study output is the provision of a Python routine for WaTEM/SEDEM applications. The application of the model is made on three different catchments, and one nested sub catchment. The authors propose to build on the idea of exploring alternative modelling attempts based on increased data availability, supporting more expert-based approaches instead of adding more complexity in existing models (as stated l. 80-83). While this idea is appealing, I have several serious concerns regarding this work:*

1. *Several papers already addressed the topic of erosion and sediment transfers in the European loess belt, and none of them was referenced. For instance, early work of Jetten et al. (1999) and Van Dijk & Kwaad (1996); Evrard et al. (2009, 2010) or recent paper such as Landemaine et al. (2023).*

2. *One strong argument of the authors is the lack of time dependent soil erosion model in literature (l. 15-16). I would strongly disagree with this hypothesis as numerous time-explicit soil erosion models exist in literature (e.g. WEPP, EROSION3D, EUROSEM, LISEM, KINEROS). The question of time-dependent variables in soil erosion modelling has also been addressed in empirical or process-based models (e.g. CREAM, SWAT, STREAM, PESERA...). How does this study built on existing approaches and why was a new methodology needed?*

3. *To enhance the model's performance, the authors surprisingly choose to include additional complexity in the model through the transport capacity parameters, which seems to contradict the paper's working hypothesis.*

4. *Modelling catchments with area in the order 100 – 1000 ha at a 15-days time step is highly questionable. It is not consistent with the time scale at which soil erosion processes are expected to occur.*

5. *It is unclear how the dataset on which the modelling approach based was processed. In particular, in such low-order catchments as the BRVL and FDTL catchments, sediment load can not be estimated from single concentrations values, due to high frequency variations in both discharge and suspended sediment concentration at the flood event scale, including hysteresis effects.*

6. *The corresponding resulting model performance is limited (NSE between 39% and 63%, mean NSE=48%), indicating that the main driving factor of the catchments erosion and sediment dynamics were not adequately captured by the proposed modelling approach.*

*I therefore not recommend publication. Please find additional comments below.*

We kindly thank the Reviewer for taking the time to review this manuscript. We appreciate the critiques and believe that based on these we can improve the quality of the manuscript. It is however disappointing that the ultimate conclusion is to not recommend the manuscript for publication. We believe that some of the main justifications for such a conclusion are based on some general misconceptions which we would like to clearly address before approaching the specific comments. Below we give responses to the general comments, and then the specific points, referencing the general justification of the manuscript (top) where necessary.

Author response 1: Firstly we thank the Reviewer for the additional recommended literature. We will rightfully take this on board within the manuscript revision and integrate the literature where possible.

Author response 2: With regards to the second point, we believe that this is a fundamental misconception of the modelling approach. We will rightly adapt the manuscript to mention the these models, most of which (e.g. WEPP, EROSION3D, EUROSEM, LISEM, KINEROS) were indeed conceptually designed to tackle the simulation of erosion at a fine spatial and resolution, and without doubt were designed with more suitability to the task of simulating the 15-day dynamics of sediment delivery. However, we used the W/S model because of its very low data input demand and evidenced continental scale application (Borrelli et al., 2017). We sought to make an application which can be used to draw comparisons with the model in its original form, which more specifically involved testing it in small nested catchments to capture the temporal variability of soil erosion and sediment delivery. Please see the general justification of the manuscript (top) in which we describe this in detail. Figure (5) in the review of de Vente et al., (2005) addresses how the aforementioned models would fall into a different category of model compared to empirically-based models such as W/S. We point the Reviewer or general reader to Merrit et al., (2003) which addresses the significant differences in model input data requirements in more detail. The data requirements of W/S make this modelling approach feasible for multiple catchments in Europe within a standardised workflow. We do not believe such an approach would be feasible for a model such as LISEM, to give an example, which would require approximately 25 different input maps.

We absolutely do not wish to contend that the question of temporally-dynamic modelling has not been addressed. We acknowledge that, amongst the > 400 distinct models or model variants which exist to simulate soil erosion in some manner (Borrelli et al., 2021), multiple models exist with a specific focus on the issue of addressing the fine-scale properties of runoff and erosion. We rather wish to contend that spatially-distributed models which are practically implementable and scalable, rarely take on

tasks involving multitemporal modelling. The review of Borrelli et al., (2021) provides a comprehensive insight into this. We note a key conclusion of Borrelli et al., (2021) on the subject of the application of process-based soil erosion models:

*"Although some models such as WEPP, RHEM, and LISEM show increasing trends of use, applications of process-based physical models appear far more constrained. Nevertheless, the scale of applications of the process-based physical models ($\tilde{x}$ = ~1 km2) suggests that the required input data are lacking for large-scale applications."*

Potential empirically-based model equivalents mentioned (e.g. SWAT), did also not meet the criteria of model selection. For example, equivalent models containing a RUSLE component contain a significant degree of spatial lumping which we considered incompatible with a fully spatially-distributed approach. In summary, our intention is not to justify on an individual basis why other models were not suitable, since we believe the general criteria of having a fully gridded spatial structure and low input data requirements justifies not using the aforementioned empirical or process-based models for this task. We believe the novelty of this study came from using a (semi-)empirical model which is widely used to measure the long-term rates of sediment redistribution and delivery at long timescales, in a temporally dynamic situation.

We will modify the manuscript to clearly outline where this modelling effort sits within the global situation of soil erosion modelling. We emphasise that this work is an example of traversing temporal scales with an existing model, rather than attempting to build a new model methodology. For example, there is no attempt in this study to create an alternative model name or frame this work as a distinct methodology.

Author response 3: Thank you for raising this valid point. We believe that only minor complexity is added within the multitemporal calibration procedure. First and foremost, we believe that the addition of calibrated parameters is justified based on the consistently poor performance of a temporally static ktc pair (ktc_high and ktc_low) within the transport capacity. Secondly, complexity is minimised by deriving the multitemporal curve via a calibrated splines curve with 6 parameters. Therefore, the W/S model is calibrated based on 6 parameters rather than its conventional 2 parameters. This calibration maintains that the multitemporal ktc pair is the only fitted parameter based on the outlet suspended sediment load data. We will add a model evaluation involving a metric which involves the trade-off between performance and model complexity (AIC, BIC, DIC etc.) to address this more explicitly.

Author response 4: Thirdly, we disagree that the temporal scale is inappropriate for the model application.  We believe the temporal scale employed is a realistic trade-off between attempting to resolve the hydrograph and sedigraph response, and employing the model in its typical long-term annual average format. Spatially distributed models which are scalable necessarily require a level of temporal aggregation, and therefore their simulation represents an integration of the time-distributed response. We considered the level of vegetation cover can be expected to be the primary control on the vulnerability of soil to rainstorm events in a humid climate. Given that it was not

feasible (or perhaps necessary) to resolve the vegetation dynamics at a finer temporal resolution in our approach, we considered that the timestep of model implementation is suitable for this particular application. We give further justifications for this within the specific responses (below) as to why a 15-day aggregation was considered the most feasible modelling timestep.

Author response 5: the catchment data is open-access to the community and used as the made publicly available by the research team. Please see Author response (8) below for a full explanation of the specific technical details relating to this point.

Author response 6: Lastly, within the justification of this study, we believe a dismissal of the study based on the NSE values is to miss the point of the study. Within such a low complexity spatially-distributed approach we accept that numerous complex controlling factors beyond those accounted for may affect the model performance. The provided NSE values in the manuscript are meant to show that the multitemporal calibration routine allows the model to fit considerably better to the 15-day variability of the sediment delivery compared to the temporally-static calibration method (in line with the study objectives). Not to suggest that the model now captures the dynamics of soil erosion and sediment delivery with a high accuracy, which would be sparsely credible for such a simplistic modelling approach.

**Specific comments**

*While model application to previously studied catchments with extensive available datasets should be a strength of this modelling study, the authors surprisingly discarded existing data. It is surprising to read that general databases were preferred over data that were specifically derived for the studied catchments. For example, plots delineation and roads network were derived from combined Integrated Administrative and Control System and Open Street Map data according to Figure 3. Why not use the specific data developed for the studied catchments, as illustrated in Matthews et al. (2023 – Figure 3) and Grangeon et al. (2022 – Figure 8) for the Kinderveld (and possibly Ganspoel), BRVL and the nested FDTL catchments (if I understood correctly data availability described in these two papers) respectively? While I understand the intention of developing a unique workflow for future applications in other catchments, it is unclear why the authors chose to discard this unique opportunity to evaluate an important source of uncertainty in input data for models, a foreseen shortcoming for future model applications on other catchments.*

*Author response 7: We point the reviewer to the general motivations of this study which we justified at the beginning of the response. We reiterate that the approach in this paper is based on optimising reproducibility across Europe rather than optimising the data inputs for a specific set of locations. The potential value of better data inputs from specific catchments is a relevant research topic, but not one addressed in this study. The specificity of W/S to its data inputs has been comprehensively evaluated (de Vente et al., 2009), and therefore our goal was not to create catchment-specific modelling situations. We addressed this comprehensively in our general justification of the method (see above) and we argue that such an approach is in no way less valuable than one with optimal data inputs.*

*Moreover, the authors did not describe how they process the raw data to establish the database used for model evaluation. The inability for the readers to evaluate how the sediment load was calculated for the BRVL and FDTL catchments is concerning, while experimental values are keys in a study intending to evaluate the benefit of a new model parameterization. I understand that 'Event-variable timestep' for the Ganspoel and Kinderveld catchments refers to the use of high frequency water height and turbidity measurements transformed into discharge and Suspended Sediment Concentration (SSC) with gauging and sampling operations (if this is correct, it should be explicitly stated). But how can Suspended Sediment Load be calculated at the runoff event scale using 'a singular aggregated sediment load'?*

Author response 8: Thank you for raising this point. We will revise the manuscript to include a better explanation of this, along with a discussion on the potential error within these time series. We chose the unit of an 'aggregated event' as the primary timestep at which to pre-process the data, before performing a secondary aggregation to the 15-day timestep at which to run the model. The aggregated event was first necessary to match the temporal structure of the RUSLE 'EI30' index which is likewise an event-scale aggregation.

This temporally-lumped 'aggregated-event' was then processed or taken directly or processed from the EUSEDcollab database. In the case of the Kinderveld and Ganspoel datasets, this was made by integrating the high frequency data into a summed value per event (done within the pre-processing of this study). Both of these datasets decompose the time series data into separate individual events (Van Oost et al., 2005). For the BRVL and FDTL datasets, the time series data is used in this study as it is made available in the EUSEDcollab database by the data producers. Specifically, each row of the data contains a timestamp for the start and end of an event, a total discharge, an average suspended sediment concentration, and a total suspended sediment load. These event-wise SSC values were calculated based on volume proportional sampling and the processing into event aggregations was done by the data providers. For this reason, "Event - aggregated " refers to a time series summarisation at the event timescale made prior to data release in EUSEDcollab. Further information on this is available in the EUSEDcollab metadata, however we will better explain these specifics for the reader.

The authors chose to decompose the 15-day dynamics of soil erosion and sediment transfers, based on the claim that 'explicit temporal dynamics are typically neglected within many soil erosion modelling approaches in favour of a focus on the long-term annual average as the predictive target' (l.16). First, this is highly questionable statement as numerous erosion and sediment transfers models exist (see, for example, the models used in the intercomparison proposed by Baartman et al., 2020; none of these models neglect temporal dynamics. One may also consider the widely used SWAT model – Arnold et al., 1998 -, which is another illustration of the inaccuracy of this assertion). Moreover, addressing the erosion dynamics of catchments in the order ~100 -1000 ha using a 15-day time step seem a large temporal window for results

aggregation relative to catchments' response time. In the end, if the model is evaluated against aggregated values, what justify this choice relative to e.g. one or several months?

Author response 9: Thank you for raising this point which we believe is quite fundamental to this modelling effort. The W/S model was conceptualised at the long-term annual average timescale. We would consider any timescale below this a downscaling exercise, in which variability would be expected to increase as the temporal scale decreases. We believe that describing the variability of sediment delivery at the 15-day timescale is an acceptably challenging task for such a model. In line with the low-complexity approach of this study, we did not attempt to predict a hydrograph or sedigraph describing the temporal concentration of runoff and suspended sediment load. Since W/S is forced using a RUSLE module, the lowest temporal unit could be the EI30 index, which is an integrated event derived from a time series of storm rainfall data. Since the individual model forcings are built on the EI30 index (i.e. active erosion periods), a 15-day aggregation will contain one (in most cases) or several events. Resampling to a 15-day timestep was needed to further overcome the differing statistical definitions of an 'event' between the model forcing data (the EI30 index) and that used for the discharge and sediment load data by the data producers (see author response 8). There is also the issue that a small EI30 value may in some cases lead to a runoff response in the channel and in other cases not, depending on antecedent conditions which are not accounted for within the model. Resampling to 15-day periods therefore allowed a standardised matching between modelled and measured sediment delivery, although we accept potential errors due to the requirement for this simplification. We justified the step of aggregating to a 15-day in lines 192-194 of the manuscript text, however we will describe in more detail the consequences of the temporal resampling process.

The model calibration procedure is unclear. As far as I understand, the model evaluation does not involved a training/testing dataset splitting, which may be a concern for adequate model evaluation. The model results are considered satisfactory, while Table 4 indicates that only the total sediment mass is adequately reproduced by the proposed modelling approach. Indeed, with a mean NSE over the four catchments of 48%, the modelled temporal dynamics can not be considered adequately simulated. This seems like an issue in a paper focusing on the improved temporal representation of a model originally developed to reproduce the total sediment mass.

Author response 10: We apologise for the difficulty in interpreting the calibration procedure. As stated in the response to Reviewer #1, we will revise the manuscript to comprehensively describe the calibration process. A comprehensive justification of the approach can be found in the general justification (above), which we intend to more clearly iterate within the manuscript. However, within the current version of the manuscript, no reference is made to the word 'satisfactory', only that:

"We show that temporal downscaling needs to be accompanied by non-linear seasonality in the transport capacity formulation to produce reasonable model performances. "

Which is a general conclusion derived from the interpretation of the relative increase in model performance of the multitemporal ktc profile when compared to those with a temporally static ktc calibration routine. Hence, although the results in Table 4 are necessary to show, we do not communicate that based on our approach that a satisfactory W/S model is now achieved to simulate the multitemporal sediment yield. In fact, the results and discussion dedicate far more attention to the interpretation of the multitemporal ktc profiles, which we believe aligned with the intentions of this manuscript. Based on this feedback we will however explicitly state this within the model text.

The authors based their study on the hypothesis that the model fails in reproducing the 'multitemporal sediment yield' because of inappropriate transport capacity parameterization, and therefore propose some improvements for the WaTEM/SEDEM model. However, in agricultural catchments, seasonality in sediment loads measured at the catchment outlet may results from variation in gross erosion. What justify the focus on improving the transport capacity? From the literature cited in the manuscript, the reader can find existing data on the studied catchment that may have been used to complement the analysis with an in-depth discussion of the relative effects of the seasonal variations of both the transport capacity and erosion rates in these agricultural catchments.

Author response 11: Thank you for this point. Transport capacity is the only calibrated parameter linking gross erosion with the output sediment load, so therefore the only interpretable parameter which was derived in the modelling processes. The RUSLE module, through its formulated factors, determines the predicted gross erosion. Lines 520-540 in the discussion address the possibility for potential seasonal biases within the RUSLE model when used for multitemporal predictions, which we believe give important insights for future improvements in the model inputs variables. Aligning with our response to reviewer #1, we intend to add a section discussing the interpretation of the seasonal differences in the ktc parameters and confront these more directly with the possibility of errors in the RUSLE gross erosion predictions.

**Technical comments**

*Table 1: Please detail how was calculated sediment load when what using a singular aggregated value per event. Please add an order of magnitude of measurements for the high frequency Ganspoel and Kinderveld catchments.*

Author response 12: As stated in Author response 8, we will add a section explaining the catchment sediment data records in more detail.

*Figure 2: It is unusual to present discharge and suspended sediment load aggregated by events. It is recommended to use more traditional time series plot including discharge,*

*suspended sediment concentration and rainfall, as it gives a significant amount of additional interesting information on e.g. stream intermittence, flood event occurrence, hysteresis, and characteristics...*

Author response 13: Thanks for the suggestion. We used aggregated events since these correspond to the temporal scale of the EI30 parameter which was the primary hydrometeorological forcing. Since we did not attempt to predict the hydrograph or sedigraph, we did not go into high temporal detail here. Moreover, the catchment literature for each catchment gives far more comprehensive insights into the aforementioned characteristics than we can provide in this study. For this reason, we contend that displaying an aggregation comparable to the predictive target of this modelling study is more appropriate. We will however add the EI30 data for each event to show the varying hydrological forcings through time.

*l.16: What characteristics are 'seasonally changing'?*

Author response 14: Thank you. We will state more clearly that we mean the vegetation cover.

*l.29: While soil crusting and vegetative boundaries (do you refer to grass strips and edges?) are recognized as important factors governing runoff dynamics, they were not explicitly studied in this paper. I would suggest removing this part from the abstract.*

Author response 15: Thank you for the comment. We will re-check the catchment literature to investigate the potential contributions of grass strips and edges during the modelled periods. Nevertheless, the point about soil crusting and vegetative boundaries was drawn from catchment observations, which we believe to be important for interpreting the model outputs. We make it as clear as possible that these conclusions are drawn from "Published catchment observations" which we believe is important to not view this modelling effort as an isolated case.

*l.140-143: I found it surprising that the extreme event recorded during the monitoring period was excluded. It is rare to find datasets including significant rainfall-runoff for analysis, as they are usually challenging to measure with an acceptable accuracy. Moreover, it is recognized that most of the catchments sediment fluxes occurred during the largest rainfall-runoff events. Such data are precious and should be analysed in details instead of being discarded.*

Author response 16: Please see Author response 14 to Reviewer #1. This was a mistake in the manuscript. In fact there was no rainfall forcing data for this event, which is why it was not included. We will correct this within the manuscript.

*l.146: susceptible to infiltration-excess.*

Author response 17: Thank you. We will correct this.

*l.175: Please define SIR.*

Author response 18: Thank you. We will do so.

*l.179-180: Should not this be written 'intensively cultivated catchments'?*

Author response 19: Yes. Thank you for noting this.

*l.192-194: It is not clear how the 15-day temporal window was defined. In particular, it seems contradictory to underline the issue of defining 'consistent thresholds of rainfall-runoff initiation when modelling discrete event episodes' and then propose a framework based on arbitrarily fixed 15-day threshold for modelling with parameters varying on a monthly basis.*

Author response 20: Thank you for this point. We will justify this step more comprehensively within the methods. We direct the Reviewer to Author response 9 for a comprehensive justification of this step.

*l.306-321: The model calibration procedure is not clear. The first paragraph refers to traditional approach in calibrating WaTEM/SEDEM, an approach discarded here, am I correct? If so, it should be removed. The second paragraph would mean that no splitting between distinct calibration/validation dataset is performed, is this correct? If so, it is a significant issue in the modelling approach.*

Author response 21: Thank you for this point. We described the typical model implementation since our intention with this modelling approach is to maintain a level of comparison between the long-term annual model implementation and the 15-day application. In the general justification (above) we justify the reasons for not splitting the dataset into a distinct calibration and validation set, in addition to how overfitting was minimised. We intend to more clearly communicate the approach, and justify why it is designed to give insights into the model rather than to perform extrapolations.

*l.324: How was defined a 'connectivity scenario'?*

Author response 22: A connectivity scenario was based on a ranking method of the different connectivity parameters into a summary index. Information on this is given in supplementary information 1.5. Please note that we intend to remove this component of the manuscript based on the justification given in Author response 35 to Reviewer #1. We will adapt the manuscript accordingly.

*l.326-328: Why not considering maximizing the NSE in the calibration procedure, which would account for both the temporality and the sediment mass?*

Author response 23: This is a valid point, however we consider it incompatible with the optimisation procedure. Specifically, because the optimisation approach is based on minimising the difference between the simulated and observed mean monthly sediment load (i.e. 12 values to represent the full timeseries), which although being a statistical feature of the full timeseries, is employed to mitigate an overfitting and saturation of the multitemporal ktc curve. This was because we aimed to account for the average seasonality, as justified in the general justification (above).

*l.362-366: The purpose of the comparison between RUSLE models and measurements is not clear here.*

Author response 24: Thank you for the comment. This was done to understand, at the catchment scale, how well the 15-day gross erosion correlated with the 15-day suspended sediment load. This allowed possibilities to infer if: 1) further processes impact the coupling between the gross erosion and the suspended sediment, or 2) significant error may exist in the temporal properties of the predicted gross erosion. Given that the W/S model with a temporally static calibration performs better when gross erosion is well correlated with the suspended sediment load, we believe this is a relevant analysis to undertake.

*l.416-419: The interpretation of relationships with low correlation coefficient is questionable. I would not recommend using three significant digits on correlation coefficient.*

Author response 25: Thank you. We will revise the number of significant digits.

*l.504-511: The introduction of pixel-based CN runoff coefficient in this study is questionable, considering that runoff was not directly evaluated in the model.*

Author response 26: This is a good point. Our goal in using the Curve Number model was to investigate if an uncalibrated runoff value could improve the modelled multitemporal sediment delivery as an out-of-the-box model to empirically estimate the runoff intensity. The reasoning behind doing so was to test the benefit of a runoff prediction which was independent of the EI30 variable within the transport capacity. Using the CN model was adjudged to be in line with a low-complexity approach. Indeed, in this study we do not attempt to model the runoff dynamics directly as part of the approach, in order to maintain a low level of complexity. The CN model was therefore not calibrated or validated based on the runoff observations, since this would have specified the model applications to the modelled catchments. Despite not calibrating or validating the CN-based runoff predictions, we did perform an evaluation in Figure (10) which we believe to be important for future model applications. Namely, that the CN model, with the input variables defined in a conventional way and incorporated into this model, tends to underestimate the runoff coefficients in the winter compared to the measured values. Apart from the other reasons mentioned (such as inducing an increased sensitivity within the W/S model), we concluded likewise that dynamic soil hydrological properties were the likely reason for the lack of improvement in the model performance. We believe that this finding is useful for other work seeking low complexity methods to incorporate runoff into a model such as W/S.

**References**

Arnold J.G., Srinivasan R., Muttiah R.S., Williams J.R. (1998). Large area hydrologic modelling and assessment part I: Model development. *Journal of the American Water Resources Association*, **31(1)**:73-89.

Baartman J. E. M., Nunes J. P., Masselink R., Darboux F., Bielders C., Degre A., Cantreul V., Cerdan O., Grangeon T., Fiener P., Wilken F., Schindewolf M., Wainwright J. (2020). What do models tell us about water and sediment connectivity? *Geomorphology*, 367, Article 107300. https://doi.org/10.1016/j.geomorph.2020.107300

Evrard O., Cerdan O., Van Wesemael B., Chauvet M., Le Bissonnais Y., Raclot D., Vandaele K., Andrieux P., Bielders C. (2009). Reliability of an expert-based runoff and erosion model: Applications of STREAM to different environments. *Catena*, **78(2):**129-141.

Evrard O., Nord G., Cerdan O., Souchère V., Le Bissonnais Y., Bonté P. (2010). Modelling the impact of land use change and rainfall seasonality on sediment export from an agricultural catchment of the northwester European loess belt. *Agriculture, Ecosystems & Environment*, **138(1-2):**83-94.

Grangeon T., Vandromme R., Pak L.T., *et al.* Dynamic parameterization of soil surface characteristics for hydrological models in agricultural catchments. *Catena* **214**, 106257 (2022). https://doi.org/10.1016/j.catena.2022.106257.

Jetten V., de Roo A., Favis-Mortlock D. (1999). Evaluation of field-scale and catchment-scale soil erosion models. *Catena*, **37:**521-541.

Landemaine V., Cerdan O., Grangeon T., Vandromme R., Laignel B., Evrard O., Salvador-Blanes S., Laceby P. (2023). Saturation-excess overland flow in the European loess belt: An underestimated process? International Soil and Water Conservation Research, **11(4):**688-699.

https://doi.org/10.1016/j.iswcr.2023.03.004

Matthews F., Verstraeten G., Borrelli P. *et al.* EUSEDcollab: a network of data from European catchments to monitor net soil erosion by water. *Sci Data* **10**, 515 (2023). https://doi.org/10.1038/s41597-023-02393-8

Van Dijk P.M., Kwaad F.J.P.M. (1996) Runoff generation and soil erosion in small agricultural catchments with loess-derived soils. *Hydrological Processes*, **10(8):**1049-1059.

**References (Authors)**

Alatorre, L. C., Beguería, S., & García-Ruiz, J. M. (2010). Regional scale modeling of hillslope sediment delivery: A case study in the Barasona Reservoir watershed (Spain) using WATEM/SEDEM. *Journal of Hydrology*, *391*(1–2), 109–123. https://doi.org/10.1016/j.jhydrol.2010.07.010

Bezak, N., Rusjan, S., Petan, S., Sodnik, J., & Mikoš, M. (2015). Estimation of soil loss by the WATEM/SEDEM model using an automatic parameter estimation procedure. *Environmental Earth Sciences*, *74*(6), 5245–5261. https://doi.org/10.1007/s12665-015-4534-0

de Vente, J., Poesen, J., Govers, G., & Boix-Fayos, C. (2009). The implications of data selection for regional erosion and sediment yield modelling. Earth Surface Processes and Landforms, 34(15), 1994–2007. https://doi.org/10.1002/ESP.1884

de Vente, J., Poesen, J., Verstraeten, G., van Rompaey, A., & Govers, G. (2008). Spatially distributed modelling of soil erosion and sediment yield at regional scales in Spain. *Global and Planetary Change*, *60*(3–4), 393–415. https://doi.org/10.1016/j.gloplacha.2007.05.002

Krause, P., Boyle, D. P., & Bäse, F. (2005). Comparison of different efficiency criteria for hydrological model assessment. *Advances in Geosciences*, *5*, 89–97. https://doi.org/10.5194/adgeo-5-89-2005

Nearing, M. A. (1998). Why soil erosion models over-predict small soil losses and under-predict large soil losses. *CATENA*, *32*(1), 15–22. https://doi.org/10.1016/S0341-8162(97)00052-0

Ke, Q., & Zhang, K. (2024). Scale issues in runoff and sediment delivery (SIRSD): A systematic review and bibliometric analysis. *Earth-Science Reviews*, *251*, 104729. https://doi.org/10.1016/J.EARSCIREV.2024.104729

Kirkby, M. J., Irvine, B. J., Jones, R. J. A., & Govers, G. (2008). The PESERA coarse scale erosion model for Europe. I. - Model rationale and implementation. European Journal of Soil Science, 59(6), 1293–1306. https://doi.org/10.1111/j.1365-2389.2008.01072.x

Peeters, I., Rommens, T., Verstraeten, G., Govers, G., van Rompaey, A., Poesen, J., & van Oost, K. (2006). Reconstructing ancient topography through erosion modelling. *Geomorphology*, *78*(3–4), 250–264. https://doi.org/10.1016/J.GEOMORPH.2006.01.033

Schmaltz, E., L. Johannsen, L., Thorsøe, M. H., Tähtikarhu, M., A. Räsänen, T., Darboux, F., & Strauss, P. (2024). Connectivity elements and mitigation measures in policy-relevant soil erosion models: A survey across Europe. *CATENA*, *234*, 107600. https://doi.org/10.1016/J.CATENA.2023.107600

Schneider, M., Schelte, T., Schmitz, F., & Körner, M. (2023). EuroCrops: The Largest Harmonized Open Crop Dataset Across the European Union. *Scientific Data*, *10*(1), 612. https://doi.org/10.1038/s41597-023-02517-0

Schmidt, S., Alewell, C., & Meusburger, K. (2018). Mapping spatio-temporal dynamics of the cover and management factor (C-factor) for grasslands in Switzerland. *Remote Sensing of Environment*, *211*(March), 89–104. https://doi.org/10.1016/j.rse.2018.04.008

Takken, I., Govers, G., Steegen, A., Nachtergaele, J., & Guérif, J. (2001). The prediction of runoff flow directions on tilled fields. *Journal of Hydrology*, *248*(1–4), 1–13. https://doi.org/10.1016/S0022-1694(01)00360-2

Verstraeten, G. (2006). Regional scale modelling of hillslope sediment delivery with SRTM elevation data. *Geomorphology*, *81*(1–2), 128–140. https://doi.org/10.1016/J.GEOMORPH.2006.04.005